# SHUFFLE PRIVATE STOCHASTIC CONVEX OPTIMIZATION

**Albert Cheu**
Georgetown University
ac2305@georgetown.edu

**Matthew Joseph**
Google Research
mtjoseph@google.com

**Jieming Mao**
Google Research
maojm@google.com

**Binghui Peng**
Columbia University
bp2601@columbia.edu

## ABSTRACT

In shuffle privacy, each user sends a collection of randomized messages to a trusted shuffler, the shuffler randomly permutes these messages, and the resulting shuffled collection of messages must satisfy differential privacy. Prior work in this model has largely focused on protocols that use a single round of communication to compute algorithmic primitives like means, histograms, and counts. We present interactive shuffle protocols for stochastic convex optimization. Our protocols rely on a new noninteractive protocol for summing vectors of bounded $\ell_2$ norm. By combining this sum subroutine with mini-batch stochastic gradient descent, accelerated gradient descent, and Nesterov's smoothing method, we obtain loss guarantees for a variety of convex loss functions that significantly improve on those of the local model and sometimes match those of the central model.

## 1 INTRODUCTION

In stochastic convex optimization, a learner receives a convex loss function $\ell \colon \Theta \times \mathcal{X} \to \mathbb{R}$ mapping pairs of parameters and data points to losses, and the goal is to use data samples $x_1, \ldots, x_n$ to find a parameter $\theta$ to minimize population loss $\mathbb{E}_{x \sim \mathcal{D}}\left[\ell(\theta, x)\right]$ over unknown data distribution $\mathcal{D}$. The general applicability of this framework has motivated a long line of work studying stochastic convex optimization problems under the constraint of differential privacy (Dwork et al., 2006b), which guarantees that the solution found reveals little information about the data used during optimization.

This has led to guarantees for both the *central* and *local* models of differential privacy. Users in central differential privacy must trust a central algorithm operator, while users in local differential privacy need not trust anyone outside their own machine. These different protections lead to different utility guarantees. When optimizing from $n$ samples over a $d$-dimensional parameter space with privacy parameter $\varepsilon$, the optimal loss term due to privacy is $O(\sqrt{d}/(\varepsilon n))$ in the central model (Bassily et al., 2013; 2019; Feldman et al., 2020). In the local model, the optimal private loss term is $O(\sqrt{d}/(\varepsilon\sqrt{n}))$ for sequentially interactive protocols (Duchi et al., 2018) and a class of compositional fully interactive protocols (Lowy and Razaviyayn, 2021).

A similar utility gap between central and local privacy also appears in many other problems (Kasiviswanathan et al., 2011; Chan et al., 2012; Duchi et al., 2018; Duchi and Rogers, 2019; Joseph et al., 2019). This has motivated the exploration of models achieving a finer trade-off between privacy and utility. One such notion of privacy is *shuffle privacy* (Cheu et al., 2019; Erlingsson et al., 2019). Here, users send randomized messages to a shuffler, which permutes the collection of messages before they are viewed by any other party. In conjunction with work on building shufflers via onion routing, multi-party computation, and secure computation enclaves, this simple model has led to a now-substantial body of work on shuffle privacy (Bittau et al., 2017; Cheu et al., 2019; Erlingsson et al., 2019; Balle et al., 2019; Ghazi et al., 2021; Balcer and Cheu, 2020; Feldman et al., 2021; Balcer et al., 2021), including steps toward real-world deployment (Google, 2021). However, comparatively little is known about shuffle private stochastic convex optimization.

## 1.1 OUR CONTRIBUTIONS

1. We introduce *sequentially interactive* (each user participates in a single protocol round) and *fully interactive* (each user participates in arbitrarily many protocol rounds) variants of the shuffle model of differential privacy[1]. This distinction is useful because, while full interactivity offers the strongest possible asymptotic guarantees, the practical obstacles to making multiple queries to a user over the duration of a protocol (Kairouz et al., 2021b; 2019) make sequentially interactive protocols more realistic.

2. We construct a noninteractive shuffle private protocol for privately computing a sum with bounded $\ell_2$ sensitivity. By using multiple messages, our protocol is substantially more accurate than existing results (see Related Work).

3. We use our $\ell_2$ summation protocol to derive several new shuffle private SCO guarantees (Table 1) via techniques including acceleration, Nesterov's smoothing method in the sequential interactive model, and noisy gradient descent in the fully interactive model.

| Loss Function | Sequential | Full |
|---|---|---|
| Convex | $O\left(\frac{d^{1/3}}{\varepsilon^{2/3}n^{2/3}}\right)$ (Thm. 4.7) | $O\left(\frac{\sqrt{d}}{\varepsilon n}\right)$ (Thm. 4.9) |
| Convex and smooth | $O\left(\frac{d^{2/5}}{\varepsilon^{4/5}n^{4/5}}\right)$ (Thm. 4.3) | |
| Strongly convex | $O\left(\frac{d^{2/3}}{\varepsilon^{4/3}n^{4/3}}\right)$ (Thm. 4.8) | $O\left(\frac{d}{\varepsilon^2 n^2}\right)$ (Thm. 4.9) |
| Strongly convex and smooth | $O\left(\frac{d}{\varepsilon^2 n^2}\right)$ (Thm. 4.8) | |

Table 1: The guarantees proved in this paper. $d$ is the dimension of the parameter space $\Theta$, $\varepsilon$ is the privacy parameter, and $n$ is the number of data points. For neatness, this table omits the non-private $1/\sqrt{n}$ term, logarithmic terms, and other parameters like Lipschitz constants, parameter space diameter, and smoothness. Full statements appear in the referenced theorems.

## 1.2 ADDITIONAL RELATED WORK

The vast majority of work on shuffle privacy studies the noninteractive model, but a few exceptions exist. Erlingsson et al. (2019) and Feldman et al. (2021) study a variant in which the shuffler is a layer that accepts randomizers from the analyst, assigns them randomly to users, and sends the permuted messages back to the analyzer. Feldman et al. (2021) also construct a basic protocol for SGD in this model. This model is similar to other work that achieves shuffle privacy guarantees by amplifying the local privacy guarantees of local randomizers (Balle et al., 2019). Erlingsson et al. (2020) also studied empirical risk minimization using shuffle-amplified sequentially interactive locally private protocols. The main difference between that work and ours is that we only require that the view of the shuffled outputs across rounds satisfies differential privacy, and can therefore avoid relying on amplification.

Another exception is the work of Beimel et al. (2020), where the shuffler broadcasts the permuted messages to all parties. Users then execute key exchange protocols, and Beimel et al. show that a first round of key exchange enables the use of multi-party computation (MPC) to simulate any central DP algorithm in the second round. Our results depart from theirs in two ways. First, we only assume that the shuffler sends outputs to the analyzer, not to individual users. Second, the MPC approach given by Beimel et al. requires an honest majority. Without an honest majority, the differential privacy guarantees of their simulation fail. In contrast, the privacy guarantees of our shuffle protocols degrade smoothly with the fraction of honest users, which can be anything larger than an arbitrary constant (see discussion at the end of Section 3.1). Simultaneous independent work by Tenenbaum et al. (2021) also studies a sequentially interactive variant of the shuffle model in the context of multi-arm bandits. Their model of sequential interactivity is identical to ours, albeit for a different problem.

---

[1]These terms are borrowed from local differential privacy (Duchi et al., 2018; Joseph et al., 2019).

Girgis et al. (2021) provide a fully interactive shuffle private protocol for empirical risk minimization (ERM). In contrast, we provide both sequentially and fully interactive protocols for SCO, which optimizes population rather than empirical loss. Their $\ell_2$ summation protocol requires one message from each user, and its second moment guarantee for the average gradient has a $1/b$ dependence on batch size $b$ (their Lemma 4). In contrast, our multi-message protocol obtains a better $1/b^2$ dependence. This is possible because our multi-message protocol does not rely on privacy amplification, and in particular circumvents their single-message lower bound (their Theorem 3). Kairouz et al. (2021b) study DP-FTRL, which uses a single pass, extends to nonconvex losses, does not rely on shuffling or amplification, and satisfies central DP. It is possible to adapt their algorithm to sequentially interactive shuffle privacy using our vector sum protocol, though their $O(d^{1/4}/\sqrt{n})$ SCO guarantee is weaker than our $O(d^{1/3}/n^{2/3})$ SCO guarantee (Theorem 4.7).

Concurrent work by Lowy and Razaviyayn (2021) study fully interactive shuffle private protocols with SCO guarantees matching ours (Section 4.2). There are two key differences between their work and ours. First, they do not provide sequentially interactive protocols. Second, their fully interactive protocols assume that users can send messages consisting of real vectors. This assumption is difficult to satisfy in practice (Canonne et al., 2020; Kairouz et al., 2021a). As a result, our protocols (and most of those in the shuffle privacy literature) rely only on discrete messages.

Finally, several shuffle private protocols are known for the simpler problem of summing scalars (Cheu et al., 2019; Balle et al., 2019; Ghazi et al., 2019; Balcer and Cheu, 2020) and recent work has studied vector addition in the secure aggregation model (Kairouz et al., 2021a), which assumes the existence of a trusted aggregator that can execute modular arithmetic on messages. We expand on this work by introducing a shuffle private protocol for summing vectors with bounded $\ell_2$ norm, with noise standard deviation proportional to the $\ell_2$ sensitivity.

## 2 PRELIMINARIES

**Differential Privacy.** Throughout, let $\mathcal{X}$ be a data universe, and suppose that each of $n$ users has one data point from $\mathcal{X}$. Two datasets $X, X' \in \mathcal{X}^n$ are *neighbors*, denoted $X \sim X'$, if they differ in the value of at most one user. Differential privacy is defined with respect to neighboring databases.

**Definition 2.1** (Dwork et al. (2006b)). *An algorithm $\mathcal{A}$ satisfies $(\varepsilon, \delta)$-differential privacy if, for every $X \sim X'$ and every event $Z$, $\mathbb{P}_{\mathcal{A}}[\mathcal{A}(X) \in Z] \leq e^\varepsilon \cdot \mathbb{P}_{\mathcal{A}}[\mathcal{A}(X') \in Z] + \delta$.*

Because Definition 2.1 assumes that the algorithm $\mathcal{A}$ has "central" access to compute on the entire raw dataset, we call this *central privacy* for brevity. For brevity, we will often use differential privacy and DP interchangeably. At times, it will also be useful to rephrase DP as a divergence constraint.

**Definition 2.2.** *The $\delta$-Approximate Max Divergence between distributions $X$ and $Y$ is*

$$D_\infty^\delta(X||Y) = \max_{Z \subseteq \text{supp}(X): \Pr[X \in Z] \geq \delta} \left[ \log \frac{\Pr[X \in Z] - \delta}{\Pr[Y \in Z]} \right].$$

**Fact 2.3.** *Algorithm $\mathcal{A}$ is $(\varepsilon, \delta)$-DP if and only if, for all $X \sim X'$, $D_\infty^\delta(\mathcal{A}(X)||\mathcal{A}(X')) \leq \varepsilon$.*

A key property of DP is closure under post-processing.

**Fact 2.4** (Proposition 2.1 (Dwork and Roth, 2014)). *Fix any function $f$. If $\mathcal{A}$ is $(\varepsilon, \delta)$-differentially private, then the composition $f \circ \mathcal{A}$ is also $(\varepsilon, \delta)$-differentially private.*

**Shuffle Privacy.** The shuffle model (Bittau et al., 2017; Cheu et al., 2019; Erlingsson et al., 2019) does not require users to trust the operator of $\mathcal{A}$. Instead, we make the narrower assumption that there is a trusted *shuffler*: users pass messages to the shuffler, the shuffler permutes them, and the algorithm $\mathcal{A}$ operates on the resulting shuffled output, thus decoupling messages from the users that sent them.

**Definition 2.5.** *A one-round shuffle protocol $\mathcal{P}$ is specified by a local randomizer $\mathcal{R} : \mathcal{X} \to \mathcal{Y}^*$ and analyzer $\mathcal{A} : \mathcal{Y}^* \to \mathcal{Z}$. In an execution of $\mathcal{P}$ on $X$, each user $i$ computes $\mathcal{R}(X_i)$—possibly using public randomness—and sends the resulting messages $y_{i,1}, y_{i,2}, \ldots$ to the shuffler $\mathcal{S}$. $\mathcal{S}$ reports $\vec{y}$ to an analyzer, where $\vec{y}$ is a uniformly random permutation of all user messages. The final output of the protocol is $\mathcal{A}(\vec{y})$.*

We now define our privacy objective in the shuffle model, often shorthanded as "shuffle privacy".

**Definition 2.6** (Cheu et al. (2019); Erlingsson et al. (2019)). $\mathcal{P} = (\mathcal{R}, \mathcal{A})$ *is* $(\varepsilon, \delta)$-*shuffle differentially private if the algorithm* $(\mathcal{S} \circ \mathcal{R}^n)(X) := \mathcal{S}(\mathcal{R}(X_1), \ldots, \mathcal{R}(X_n))$, *i.e. the analyzer's view of the shuffled messages, is* $(\varepsilon, \delta)$-*differentially private. The privacy guarantee is only over the internal randomness of the users' randomizers and the shuffler.*

A drawback of the preceding definition of shuffle protocols is that it limits communication to one round. It is not possible, for example, to adjust the randomizer assigned to user $i$ based on the messages reported from user $i - 1$. This is an obstacle to implementing adaptive algorithms like gradient descent. We therefore extend the shuffle model with the following *sequentially interactive* and *fully interactive* variants. Sequential interactivity breaks the data into batches and runs a different shuffle protocol on each batch in (possibly adaptive) sequence.

**Definition 2.7.** *Let* Tr *denote the universe of shuffle protocol transcripts, and let* $\mathfrak{R}$ *denote the universe of randomizers. A sequentially interactive shuffle protocol* $\mathcal{P}$ *consists of initial local randomizer* $\mathcal{R}_0$, *analyzer* $\mathcal{A}$, *and update function* $U \colon \mathsf{Tr} \to [n] \times \mathfrak{R}$. *Let* $\mathsf{Tr}_t$ *denote the transcript after* $t$ *rounds of the protocol. In each round* $t$ *of* $\mathcal{P}$, *the analyzer computes* $(n_t, \mathcal{R}_t) = U(\mathsf{Tr}_{t-1})$, $n_t$ *new users apply* $\mathcal{R}_t$ *to their data points, and* $\mathcal{S}$ *shuffles the result and relays it back to* $\mathcal{A}$. *At the conclusion of a* $T$-*round protocol, the analyzer releases final output* $\mathcal{A}(\mathsf{Tr}_T)$.

This formalizes the idea that the analyzer looks at the past shuffled outputs to determine how many and what kind of randomizers to pass to the next batch of users. Note that, in a sequentially interactive protocol, each user only participates in one computation. More generally, we can further allow the analyzer to repeatedly query users. This leads us to the *fully interactive model*.

**Definition 2.8.** *A fully interactive shuffle protocol* $\mathcal{P}$ *is identical to a sequentially interactive shuffle protocol, except the update function* $U \colon \mathsf{Tr} \to 2^{[n]} \times \mathfrak{R}$ *selects an arbitrary subset of users in each round. In particular, a user may participate in multiple rounds.*

The main advantage of full interactivity is that it offers the strongest formal utility guarantees (see Section 4); the main advantage of sequential interactivity is that it is typically difficult to query a user multiple times in practice (Kairouz et al., 2021b; 2019).

For sequentially and fully interactive shuffle protocols, the definition of privacy is identical: the view of the transcript satisfies DP. The noninteractive definition is a special case of this general definition.

**Definition 2.9.** *Given shuffle protocol* $\mathcal{P}$, *let* $\mathcal{M}_{\mathcal{P}} \colon \mathcal{X}^* \to \mathsf{Tr}$ *denote the central algorithm that simulates* $\mathcal{P}$ *on an input database and outputs the resulting transcript. Then* $\mathcal{P}$ *is* $(\varepsilon, \delta)$-*shuffle differentially private if* $\mathcal{M}_{\mathcal{P}}$ *is* $(\varepsilon, \delta)$-*differentially private.*

**Stochastic Convex Optimization.** Let closed convex $\Theta \subset \mathbb{R}^d$ with diameter $D$ be our parameter space. We always assume loss functions $\ell(\theta, x)$ that are convex and $L$-Lipschitz in $\Theta$.

**Definition 2.10.** *Loss function* $\ell \colon \Theta \times \mathcal{X} \to \mathbb{R}$ *is* convex *in* $\Theta$ *if, for all* $x \in \mathcal{X}$, *for all* $t \in [0, 1]$ *and* $\theta, \theta' \in \Theta$, $\ell(t\theta + (1-t)\theta', x) \leq t\ell(\theta, x) + (1-t)\ell(\theta', x)$. $\ell$ *is* $L$-Lipschitz *in* $\Theta$ *if for all* $x \in \mathcal{X}$ *and* $\theta, \theta' \in \Theta$, $|\ell(\theta, x) - \ell(\theta', x)| \leq L\|\theta - \theta'\|_2$.

In some cases, we also assume that our loss function $\ell$ is $\beta$-smooth and/or $\lambda$-strongly convex.

**Definition 2.11.** *Loss function* $\ell \colon \Theta \times \mathcal{X} \to \mathbb{R}$ *is* $\beta$-smooth *over* $\Theta$ *if, for every* $x \in \mathcal{X}$ *and for every* $\theta, \theta' \in \Theta$, $\|\nabla\ell(\theta, x) - \nabla\ell(\theta', x)\|_2 \leq \beta\|\theta - \theta'\|_2$. $\ell$ *is* $\lambda$-strongly convex *in* $\Theta$ *if, for all* $x \in \mathcal{X}$, *for all* $t \in [0, 1]$ *and* $\theta, \theta' \in \Theta$, $\ell(t\theta + (1-t)\theta', x) \leq t\ell(\theta, x) + (1-t)\ell(\theta', x) - \frac{\lambda t(1-t)}{2}\|\theta - \theta'\|_2^2$.

Our goal is to minimize population loss. We view each user as a sample from $\mathcal{D}$, and our protocols learns through repeated interaction with the shuffler rather than direct access to the samples.

**Definition 2.12.** *Let* $\mathcal{D}$ *be a distribution over* $\mathcal{X}$, *and let* $\ell \colon \Theta \times \mathcal{X} \to \mathbb{R}$ *be a loss function. Then algorithm* $\mathcal{A} \colon \mathcal{X}^* \to \Theta$ *has* population loss $\mathbb{E}_{\mathcal{A}, \mathcal{D}}\left[\ell(\bar{\theta}, \mathcal{D})\right] = \mathbb{E}_{\theta \sim \mathcal{A}(\mathcal{D}^n)}\left[\mathbb{E}_{x \sim \mathcal{D}}\left[\ell(\theta, x)\right]\right]$.

## 3 VECTOR SUMMATION

In this section, we provide a shuffle private protocol $\mathcal{P}_{\text{VEC}}$ for summing vectors of bounded $\ell_2$ norm. This contrasts with existing work, which focuses on $\ell_1$ sensitivity. $\mathcal{P}_{\text{VEC}}$ will be useful for the gradient descent steps of our algorithms in Section 4. $\mathcal{P}_{\text{VEC}}$ relies on $d$ invocations of a scalar sum protocol

$\mathcal{P}_{1D}$, one for each dimension. We describe this scalar summation subroutine in Section 3.1, then apply it to vector summation in Section 3.2. Proofs for results in this section appear in Appendix A

### 3.1 SCALAR SUM SUBROUTINE

We start with the pseudocode for $\mathcal{P}_{1D}$, a shuffle protocol for summing scalars. As is generally the case for shuffle protocols, the overall protocol decomposes $\mathcal{P}_{1D} = (\mathcal{R}_{1D}, \mathcal{A}_{1D})$ into randomizer and analyzer components: each user $i$ computes a collection of messages $\vec{y}_i \sim \mathcal{R}_{1D}(x_i)$, the shuffler $\mathcal{S}$ aggregates and permutes these messages to produce $\vec{y} \in \{0,1\}^{(g+b)n}$, and the analyzer takes the result as input for its final analysis $\mathcal{A}_{1D}(\vec{y})$. $\mathcal{P}_{1D}$ uses the fixed-point encoding presented by Cheu et al. (2019) and ensures privacy using a generalization of work by Balcer and Cheu (2020).

---

**Algorithm 1** $\mathcal{P}_{1D}$, a shuffle protocol for summing scalars

1: **Parameters:** Scalar database $X = (x_1, \ldots, x_n) \in [0, \Delta]^n$; $g, b \in \mathbb{N}$; $p \in (0, 1/2)$
2: **procedure** LOCAL RANDOMIZER $\mathcal{R}_{1D}(x)$
3:      Set $\overline{x} \leftarrow \lfloor xg/\Delta \rfloor$
4:      Sample rounding value $\eta_1 \sim \mathbf{Ber}(xg/\Delta - \overline{x})$
5:      Set $\hat{x} \leftarrow \overline{x} + \eta_1$
6:      Sample privacy noise value $\eta_2 \sim \mathbf{Bin}(b, p)$
7:      Report $\hat{x} + \eta_2$ copies of 1 and $g + b - (\hat{x} + \eta_2)$ copies of 0
8: **end procedure**
9: **procedure** ANALYZER $\mathcal{A}_{1D}(\vec{y})$
10:      Output estimator $\frac{\Delta}{g}\left(\left(\sum_{i=1}^{(g+b)n} y_i\right) - pbn\right)$
11: **end procedure**

---

$\mathcal{P}_{1D}$'s privacy guarantee is instance-specific: for every pair of inputs, we bound the approximate max-divergence between output distributions as a function of the change in inputs. Previously proposed by Chatzikokolakis et al. (2013), this property will be essential when proving that $\mathcal{P}_{VEC}$ is private.

**Lemma 3.1.** *Fix any number of users $n$, $\varepsilon \leq 15$, and $0 < \delta < 1/2$. Let $g \geq \Delta\sqrt{n}$, $b > \frac{180g^2 \ln(2/\delta)}{\varepsilon^2 n}$, and $p = \frac{90g^2 \ln(2/\delta)}{b\varepsilon^2 n}$. Then 1) for any neighboring databases $X \sim X' \in [0, \Delta]^n$ that differ on user $u$, $D_\infty^\delta((\mathcal{S} \circ \mathcal{R}_{1D}^n)(X) || (\mathcal{S} \circ \mathcal{R}_{1D}^n)(X')) \leq \varepsilon \cdot \left(\frac{2}{g} + \frac{|x_u - x_u'|}{\Delta}\right)$, and 2) for any input database $X \in [0, \Delta]^n$, $\mathcal{P}_{1D}(X)$ is an unbiased estimate of $\sum_{i=1}^n x_i$ and has variance $O(\frac{\Delta^2}{\varepsilon^2} \log \frac{1}{\delta})$.*

We defer the proof to Appendix A. Claim 1 follows first from the observation that the adversary's view is equivalent to the sum of the message bits. This sum has binomial noise, which was first analyzed by Dwork et al. (2006a) in the context of central DP. We adapt work on a shuffle private variant due to Ghazi et al. (2021) to incorporate a dependence on the per-instance distance between databases. Claim 2 follows as a straightforward calculation.

Lemma 3.1 holds when all $n$ users follow the protocol. We call these users *honest*. It is possible to prove a more general version of Lemma 3.1 whose parameters smoothly degrade with the fraction of honest users (see Appendix A.3). This *robust* variant of shuffle privacy was originally defined in work by Balcer et al. (2021) and naturally extends to $\mathcal{P}_{VEC}$.

The remaining analysis uses $\mathcal{P}_{1D}$ in a black-box manner. It is therefore possible to replace it with a lower-communication subroutine for scalar sum (Balle et al., 2019). However, this improved communication requires a more complicated protocol with an additional subroutine. Our work focuses on sample complexity guarantees, so we use $\mathcal{P}_{1D}$ for clarity. Nonetheless, we note that each user sends $g + b \approx \Delta\sqrt{n} + \Delta^2 \log(1/\delta)/\varepsilon^2$ bits in $\mathcal{P}_{1D}$, and the analyzer processes these in time $O((g+b)n)$. Scaling these quantities by $d$ yields guarantees for $\mathcal{P}_{VEC}$ in the next subsection. Moreover, it is possible to remove the communication dependence on $\Delta$ by having users scale down their inputs by $\Delta$ before running $\mathcal{P}_{1D}$, running $\mathcal{P}_{1D}$ as if $\Delta = 1$, and then scaling up the final output by $\Delta$ to compensate; this does not affect the overall unbiasedness or variance guarantees. Improving the communication efficiency of these protocols may be an interesting direction for future work.

## 3.2 VECTOR SUM

Below, we present the pseudocode for $\mathcal{P}_{\text{VEC}}$, again decomposing into randomizer and analyzer components. Note that we view the vector $\vec{y}$ to be the collection of all shuffled messages and, since the randomizers labels these messages by coordinate, $\vec{y}_j$ consists of the messages labelled $j$. The accompanying guarantee for $\mathcal{P}_{\text{VEC}}$ appears in Theorem 3.2.

---

**Algorithm 2** $\mathcal{P}_{\text{VEC}}$, a shuffle protocol for vector summation

---

1: **Input:** database of $d$-dimensional vectors $\vec{X} = (\vec{x}_1, \cdots, \vec{x}_n)$; privacy parameters $\varepsilon, \delta; \Delta_2$.
2: **procedure** LOCAL RANDOMIZER $\mathcal{R}_{\text{VEC}}(\vec{x}_i)$
3:     **for** coordinate $j \in [d]$ **do**
4:         Shift data to enforce non-negativity: $\vec{w}_{i,j} \leftarrow \vec{x}_{i,j} + (\Delta_2, \ldots, \Delta_2)$
5:         $\vec{m}_j \leftarrow \mathcal{R}_{\text{1D}}(\vec{w}_{i,j})$
6:     **end for**
7:     Output labeled messages $\{(j, \vec{m}_j)\}_{j \in [d]}$
8: **end procedure**
9: **procedure** ANALYZER $\mathcal{A}_{\text{VEC}}(\vec{y})$
10:     **for** coordinate $j \in [d]$ **do**
11:         Run analyzer on $j$'s messages $z_j \leftarrow \mathcal{A}_{\text{1D}}(j, \vec{y}_j)$
12:         Re-center: $o_j \leftarrow z_j - n\Delta_2$
13:     **end for**
14:     Output the vector of estimates $\vec{o}$
15: **end procedure**

---

**Theorem 3.2.** *For any $0 < \varepsilon \le 15$, $0 < \delta < 1/2$, $d, n \in \mathbb{N}$, and $\Delta_2 > 0$, there are choices of parameters $b, g \in \mathbb{N}$ and $p \in (0, 1/2)$ for $\mathcal{P}_{\text{1D}}$ (Algorithm 3.1) such that, for inputs $\vec{X} = (\vec{x}_1, \ldots, \vec{x}_n)$ of vectors with maximum norm $||\vec{x}_i||_2 \le \Delta_2$, 1) $\mathcal{P}_{\text{VEC}}$ is $(\varepsilon, \delta)$-shuffle private and, 2) $\mathcal{P}_{\text{VEC}}(\vec{X})$ is an unbiased estimate of $\sum_{i=1}^{n} \vec{x}_i$ and has bounded variance*

$$\mathbb{E}\left[\left\|\mathcal{P}_{\text{VEC}}(\vec{X}) - \sum_{i=1}^{n} \vec{x}_i\right\|_2^2\right] = O\left(\frac{d\Delta_2^2}{\varepsilon^2}\log^2\frac{d}{\delta}\right).$$

The proof relies on a variant of advanced composition (Lemma 3.3). Although the proof follows many of the same steps as prior work, the statement is somewhat nonstandard because it uses the divergence between two specific algorithm executions, rather than the worst-case divergence of a generic differential privacy guarantee. This enables us to use Lemma 3.1.

**Lemma 3.3.** *Fix any $\gamma \in (0, 1)$ and neighboring databases $X, X' \in \mathcal{X}^n$. For all $j \in [d]$, suppose algorithm $\mathcal{A}_j : \mathcal{X}^n \to \mathcal{Y}$ use independent random bits and satisfies $D_\infty^{\delta_j}(\mathcal{A}_j(X)||\mathcal{A}_j(X')) \le \varepsilon_j$. Let*

$$\varepsilon' = \sum_{j=1}^{d} \varepsilon_j(e^{\varepsilon_j} - 1) + 2\sqrt{\log(1/\gamma)\sum_{j=1}^{d}\varepsilon_j^2} \quad \text{and} \quad \delta' = \sum_{j=1}^{d}\delta_j + \gamma.$$

*Then $D_\infty^{\delta'}(\mathcal{A}_1(X), \ldots, \mathcal{A}_d(X)||\mathcal{A}_1(X')\ldots, \mathcal{A}_d(X')) \le \varepsilon'$.*

We defer the proof of Theorem 3.2 to Appendix A.2 but sketch the outline here. For any neighboring pair of datasets, the $j$-th invocation of $\mathcal{P}_{\text{1D}}$ by $\mathcal{P}_{\text{VEC}}$ produces one of two message distributions such that the divergence between them is roughly proportional to the gap in the $j$-th coordinate of the input. This is immediate from Lemma 3.1. The fact that we know a bound on sensitivity allows us to bound the (squared) $\ell_2$ norm of the vector of divergences. This norm is implicit within Lemma 3.3, so the target theorem follows by re-scaling of parameters and substitution.

## 4 CONVEX OPTIMIZATION

We now apply $\mathcal{P}_{\text{VEC}}$ to convex optimization. Section 4.1 describes shuffle protocols that are sequentially interactive and solicit one input from each user.[2] Section 4.2 offers stronger utility guarantees

---

[2]Since sequentially interactive protocols process data online, our sequentially interactive shuffle protocols also have straightforward *pan-private* (Dwork et al., 2010a) analogues with the same guarantees; see Appendix C.

via fully interactive protocols that query users multiple times. Complete proofs of all results in this section, along with communication and runtime guarantees, appear in Appendix B.

## 4.1 SEQUENTIALLY INTERACTIVE PROTOCOLS

Our first sequentially interactive protocol is a simple instantiation of noisy mini-batch SGD, $\mathcal{P}_{\mathrm{SGD}}$. This baseline algorithm achieves population loss $O(d^{1/4}/\sqrt{\varepsilon n})$ (Theorem 4.1), which already improves on the locally private guarantee. However, in a departure from the central and local models, we then show that a still better $O(d^{1/3}/(\varepsilon n)^{2/3})$ loss guarantee is possible (Theorem 4.7) using a more complex algorithm $\mathcal{P}_{\mathrm{AGD}}$ that smooths the loss function and employs accelerated SGD.

**Convex $\ell$** We start with $\mathcal{P}_{\mathrm{SGD}}$. Because $\mathcal{P}_{\mathrm{SGD}}$ is sequentially interactive, for a fixed number of users $n$, the batch size $b$ is inversely proportional to the number of iterations $T$. As a result, Theorem 4.1 chooses $b$ to balance the noise added to each gradient step (which decreases with $b$) with the number of steps taken (which also decreases with $b$).

---

**Algorithm 3** $\mathcal{P}_{\mathrm{SGD}}$, Sequentially interactive shuffle private SGD

---

**Require:** Number of users $n$, batch size $b$, privacy parameter $\varepsilon, \delta$, step size $\eta$, Lipschitz parameter $L$
1: Initialize parameter estimate $\theta_0 \in \Theta$, and set number of iterations $T = n/b$
2: **for** $t = 1, 2, \ldots, T$ **do**
3:     Compute gradient at $\theta_{t-1}$, $\bar{g}_t \leftarrow \frac{1}{b}\mathcal{P}_{\mathrm{VEC}}\left(\nabla_\theta \ell(\theta_{t-1}, x_{t,1}), \cdots, \nabla_\theta \ell(\theta_{t-1}, x_{t,b}); \varepsilon, \delta, L\right)$
4:     Update parameter estimate $\theta_t \leftarrow \pi_\Theta\left(\theta_{t-1} + \eta \bar{g}_t\right)$
5: **end for**
6: Output $\frac{1}{T}\sum_{t=0}^{T-1}\theta_t$

---

**Theorem 4.1.** *Let loss function $\ell$ be convex, $L$-Lipschitz over a closed convex set $\Theta \subset \mathbb{R}^d$ of diameter $D$, $\bar{\theta} = \frac{1}{T}\sum_{t=0}^{T-1}\theta_t$, $b = \frac{\sqrt{d}\log(d/\delta)}{\varepsilon}$, $T = n/b$, $\eta = \frac{\varepsilon b D}{L\sqrt{T}(\varepsilon b + \sqrt{d}L\log(d/\delta))}$, and $\varepsilon \leq 15$. Then $\mathcal{P}_{\mathrm{SGD}}$ is $(\varepsilon, \delta)$-shuffle private and has population loss*

$$\mathbb{E}_{\mathcal{P}_{\mathrm{SGD}}, \mathcal{D}}\left[\ell(\bar{\theta}, \mathcal{D})\right] \leq \min_{\theta \in \Theta} \ell(\theta, \mathcal{D}) + O\left(\frac{d^{1/4}DL\log^{1/2}(d/\delta)}{\sqrt{\varepsilon n}}\right).$$

Privacy—for $\mathcal{P}_{\mathrm{SGD}}$ and our other protocols, which all rely on $\mathcal{P}_{\mathrm{VEC}}$—follows from the fact that the $L$-Lipschitz assumption ensures that a loss gradient has $\ell_2$ norm at most $L$. The utility proof proceeds by viewing each noisy gradient step as a call to a noisy gradient oracle. This enables us to use a standard result for gradient oracle SGD.

**Lemma 4.2** (Bubeck et al. (2015)). *Suppose $\mathcal{P}_{\mathrm{SGD}}$ queries an $L_G$-noisy gradient oracle at each iteration, then its output satisfies $\mathbb{E}\left[\ell(\bar{\theta}, \mathcal{D})\right] \leq \min_{\theta \in \Theta} \ell(\theta, \mathcal{D}) + \frac{D^2}{2\eta T} + \frac{\eta L_G^2}{2}$.*

In our case, we can bound the noise level $L_G$ as a function of our problem and privacy parameters, including the batch size. We then choose the batch size to minimize the overall loss guarantee.

**Convex and smooth $\ell$** Next, we obtain a better loss guarantee than Theorem 4.1 when the loss function is also smooth. The key insight is that by using acceleration, a better trade-off between noise level and number of iterations is possible. The resulting algorithm, $\mathcal{P}_{\mathrm{AGD}}$, can be seen as a private version of the accelerated stochastic approximation algorithm (AC-SA) of Lan (2012).

**Theorem 4.3.** *Let $\ell$ be convex, $\beta$-smooth, and $L$-Lipschitz over closed convex set $\Theta \subset \mathbb{R}^d$ of diameter $D$. Denote $\bar{\theta} = \theta_{t+1}^{\mathrm{ag}}$. Let batch size $b = \frac{n^{3/5}d^{1/5}L^{2/5}\log^{2/5}(d/\delta)}{\varepsilon^{2/5}\beta^{2/5}D^{2/5}}$, $L_t = \frac{1}{t+1}((T+2)^{3/2}\frac{\sigma}{D} + \beta)$, $\alpha_t = \frac{2}{t+2}$, and $\varepsilon \leq 15$. Then $\mathcal{P}_{\mathrm{AGD}}$ is $(\varepsilon, \delta)$-shuffle private with population loss*

$$\mathbb{E}_{\mathcal{P}_{\mathrm{AGD}}, \mathcal{D}}\left[\ell(\bar{\theta}, \mathcal{D})\right] \leq \min_{\theta \in \Theta} \ell(\theta, \mathcal{D}) + O\left(\frac{DL}{\sqrt{n}} + \frac{d^{2/5}\beta^{1/5}D^{6/5}L^{4/5}\log^{4/5}(d/\delta)}{\varepsilon^{4/5}n^{4/5}}\right).$$

The proof of Theorem 4.3 is similar to that of Theorem 4.1. The main difference is that we rely on an oracle utility guarantee that is specific to accelerated gradient descent:

---

**Algorithm 4** $\mathcal{P}_{\text{AGD}}$, Sequentially interactive shuffle private AC-SA

---

**Require:** Number of users $n$, batch size $b$, privacy parameters $\varepsilon, \delta$, Lipchitz parameter $L$, learning rate sequence $\{L_t\}, \{\alpha_t\}$

1: Initialize parameter estimate $\theta_1^{\text{ag}} = \theta_1 \in \Theta$, and set number of iterations $T = \lfloor n/b \rfloor$
2: **for** $t = 1, 2, \ldots, T$ **do**
3:      Update middle parameter estimate $\theta_t^{\text{md}} \leftarrow \alpha_t \theta_t + (1 - \alpha_t)\theta_t^{\text{ag}}$.
4:      Estimate gradient at $\theta_t^{\text{md}}$, $\bar{g}_t \leftarrow \frac{1}{b}\mathcal{P}_{\text{VEC}}\left(\nabla_\theta \ell(\theta_t^{\text{md}}, x_{t,1}), \cdots, \nabla_\theta \ell(\theta_t^{\text{md}}, x_{t,b}); \varepsilon, \delta, L\right)$
5:      Update parameter estimate $\theta_{t+1} \leftarrow \arg\min_{\theta \in \Theta}\left\{\langle \bar{g}_t, \theta - \theta_t \rangle + \frac{L_t}{2}\|\theta - \theta_t\|_2^2\right\}$
6:      Update aggregated parameter estimate $\theta_{t+1}^{\text{ag}} \leftarrow \alpha_t \theta_{t+1} + (1 - \alpha_t)\theta_t^{\text{ag}}$
7: **end for**
8: Output $\theta_{T+1}^{\text{ag}}$

---

**Lemma 4.4** (Theorem 2 (Lan, 2012)). *Suppose $\mathcal{P}_{\text{AGD}}$ receives a noisy gradient oracle with variance (at most) $\sigma$ in each iteration. Taking $L_t = \frac{1}{t+1}((T+2)^{3/2}\frac{\sigma}{D} + \beta), \alpha_t = \frac{2}{t+2}$, then the expected error of $\mathcal{P}_{\text{AGD}}$ can be bounded as*

$$\mathbb{E}\left[\ell(\bar{\theta}, \mathcal{D})\right] \leq \min_{\theta \in \Theta} \ell(\theta, \mathcal{D}) + O\left(\frac{\beta D^2}{T^2} + \frac{D\sigma}{\sqrt{T}}\right).$$

**Smoothing for convex non-smooth $\ell$**    Perhaps surprisingly, when the target loss function is convex and non-smooth, the SGD approach of algorithm $\mathcal{P}_{\text{SGD}}$ is suboptimal. We improve the convergence rate by combining the accelerated gradient descent method $\mathcal{P}_{\text{AGD}}$ with Nesterov's smoothing technique. In particular, for a non-smooth loss function, we approximate it with a smooth function by exploiting Moreau-Yosida regularization. We then optimize the smooth function via $\mathcal{P}_{\text{AGD}}$. This is not the first application of Moreau-Yosida regularization in the private optimization literature (Bassily et al., 2019), but we depart from past work by combining it with acceleration to attain a better guarantee. We start by recalling the notion of the $\beta$-Moreau envelope, using the language of Bassily et al. (2019).

**Definition 4.5.** *For $\beta > 0$ and convex $f: \theta \to \mathbb{R}^d$, the $\beta$-Moreau envelope $f_\beta: \theta \to \mathbb{R}^d$ of $f$ is*

$$f_\beta(\theta) = \min_{\theta' \in \Theta}\left(f(\theta') + \frac{\beta}{2}\|\theta - \theta'\|_2^2\right).$$

The following properties of the $\beta$-Moreau envelope will be useful.

**Lemma 4.6** (Nesterov (2005)). *Let $f: \theta \to \mathbb{R}^d$ be a convex function, and define the proximal operator $\text{prox}_f(\theta) = \arg\min_{\theta' \in \Theta}\left(f(\theta') + \frac{1}{2}\|\theta - \theta'\|_2^2\right)$. For $\beta > 0$:*

1. *$f_\beta$ is convex, $2L$-Lipschitz and $\beta$-smooth.*

2. *$\forall \theta \in \Theta$, $\nabla f_\beta(\theta) = \beta(\theta - \text{prox}_{f/\beta}(\theta))$.*

3. *$\forall \theta \in \Theta$, $f_\beta(\theta) \leq f(\theta) \leq f_\beta(\theta) + \frac{L^2}{2\beta}$.*

Informally, we use Lemma 4.6 as follows. First, claim 1 enables us to optimize $f_\beta$ using $\mathcal{P}_{\text{AGD}}$. Inside $\mathcal{P}_{\text{AGD}}$, we use claim 2 to compute the desired gradients of $f_\beta$ and obtain a loss guarantee for $f_\beta$ using Theorem 4.3. Finally, claim 3 bounds the change in loss between $f_\beta$ and the true function $f$ as a function of $\beta$. We then choose $\beta$ to minimize the overall loss guarantee.

**Theorem 4.7.** *Let $\ell$ be convex and $L$-Lipschitz over a closed convex set $\Theta \subset \mathbb{R}^d$ of diameter $D$ and $\varepsilon \leq 15$. Then combining $\mathcal{P}_{\text{AGD}}$ with the above smoothing method yields an $(\varepsilon, \delta)$-shuffle private algorithm with population loss*

$$\mathbb{E}\left[\ell(\bar{\theta}, \mathcal{D})\right] \leq \min_{\theta \in \Theta} \ell(\theta, \mathcal{D}) + O\left(\frac{DL}{\sqrt{n}} + \frac{d^{1/3}DL\log^{2/3}(d/\delta)}{\varepsilon^{2/3}n^{2/3}}\right).$$

**Strongly convex function**    When the function is strongly convex, we use a folklore reduction from the convex setting: start from arbitrary point $\theta_0 \in \Theta$, and apply a convex optimization algorithm

$k = O(\log \log n)$ times, where the $j$-th ($j \in [k]$) application uses the output from previous phase as the initial point and $n_j = n/k$ samples. In the strongly convex but possibly non-smooth case, we apply the smoothing version of $\mathcal{P}_{\mathrm{AGD}}$ used in Theorem 4.7. In the strongly convex and smooth case, we apply the version of $\mathcal{P}_{\mathrm{AGD}}$ used in Theorem 4.3. A similar approach appears in the work of Feldman et al. (2020); the adaptation for our setting requires, among other modifications, a more careful analysis of the optimization trajectory.

**Theorem 4.8.** *Let $\ell$ be $\lambda$-strongly convex and $L$-Lipschitz over a closed convex set $\Theta \subset \mathbb{R}^d$ of diameter $D$ and $\varepsilon \leq 15$. Then combining $\mathcal{P}_{\mathrm{AGD}}$ with the reduction above yields an $(\varepsilon, \delta)$-shuffle private algorithm with population loss*

$$\mathbb{E}\left[\ell(\bar{\theta}, \mathcal{D})\right] \leq \min_{\theta \in \Theta} \ell(\theta, \mathcal{D}) + \tilde{O}\left(\frac{L^2}{\lambda n} + \frac{d^{2/3} L^2 \log^{4/3}(d/\delta)}{\lambda \varepsilon^{4/3} n^{4/3}}\right).$$

*If the loss function is also $\beta$-smooth, then the population loss can be improved to*

$$\mathbb{E}\left[\ell(\bar{\theta}, \mathcal{D})\right] \leq \min_{\theta \in \Theta} \ell(\theta, \mathcal{D}) + \tilde{O}\left(\frac{L^2}{\lambda n} + \frac{d \beta^{1/2} L^2 \log^2(d/\delta)}{\lambda^{3/2} \varepsilon^2 n^2}\right).$$

## 4.2 Fully Interactive Protocol

We conclude with fully interactive protocols, which allow a user to participate in multiple shuffles. In this expanded setting, a version of $\mathcal{P}_{\mathrm{SGD}}$ obtains loss guarantees that match those of the *central* model of DP, up to logarithmic factors. The pseudocode of our protocol $\mathcal{P}_{\mathrm{GD}}$ is shown in Algorithm 5. For strongly convex functions, we further divide users into disjoint groups and apply a fully interactive protocol $\mathcal{P}_{\mathrm{GD}}$ to each group in sequence, using the parameter estimate learned from one group as the initial parameter estimate for the next.

---

**Algorithm 5** $\mathcal{P}_{\mathrm{GD}}$, Fully interactive shuffle private gradient descent

---

**Require:** Number of users $n$, privacy parameters $\varepsilon, \delta$, Lipschitz parameter $L$, number of iterations $T$, step size $\eta$
  1: Initialize parameter estimate $\theta_0 \leftarrow 0$
  2: **for** $t = 1, 2, \ldots, T$ **do**
  3:     Compute the gradient at $\theta_t$,

$$\bar{g}_t \leftarrow \frac{1}{n} \mathcal{P}_{\mathrm{VEC}}\left(\nabla_\theta \ell(\theta_t, x_1), \cdots, \nabla_\theta \ell(\theta_t, x_n); \frac{\varepsilon}{2\sqrt{2T \log(1/\delta)}}, \frac{\delta}{T+1}, L\right)$$

  4:     Compute and store the parameter $\theta_{t+1} \leftarrow \pi_\Theta\left(\theta_t + \eta \bar{g}_t\right)$
  5: **end for**
  6: Output final estimate $\bar{\theta} \leftarrow \frac{1}{T} \sum_{t=0}^{T-1} \theta_t$

---

**Theorem 4.9.** *Let $\ell$ be convex and $L$-Lipschitz over a closed convex set $\Theta \subset \mathbb{R}^d$ of diameter $D$ and $\varepsilon \leq 15$. Then the protocol $\mathcal{P}_{\mathrm{GD}}$ is $(\varepsilon, \delta)$-shuffle private with population loss*

$$\mathbb{E}\left[\ell(\bar{\theta}, \mathcal{D})\right] \leq \min_{\theta \in \Theta} \ell(\theta, \mathcal{D}) + O\left(\frac{DL}{\sqrt{n}} + \frac{d^{1/2} DL \log^{3/2}(d/\delta)}{\varepsilon n}\right).$$

*If $\ell$ is $\lambda$-strongly convex, we can further improve the population loss to*

$$\mathbb{E}\left[\ell(\bar{\theta}, \mathcal{D})\right] \leq \min_{\theta \in \Theta} \ell(\theta, \mathcal{D}) + O\left(\frac{L^2}{\lambda}\left[\frac{1}{n} + \frac{d \log^3(d/\delta)}{\varepsilon^2 n^2}\right]\right).$$

Details appear at the end of Appendix B; since the algorithm and analysis combines previously-discussed components, we sketch them here. The analysis employs the same Moreau envelope approach used in Theorem 4.7. However, unlike in the analysis of Theorem 4.7, we no longer require acceleration. Advanced composition and an appropriate setting of the parameters completes the result. For strongly convex function, we need to further split users into $\log \log n$ disjoint groups and apply $\mathcal{P}_{\mathrm{GD}}$ to these disjoint groups of users in sequence.

ACKNOWLEDGEMENTS

We thank the anonymous ICLR reviewers for their helpful comments. Binghui Peng is supported by Christos Papadimitriou's NSF grant CCF-1763970 AF, CCF-1910700 AF, DMS-2134059 and a softbank grant, and by Xi Chen's NSF grant CCF-1703925 and IIS-1838154. Albert Cheu is supported in part by a gift to Georgetown University. Part of his work was done while he was a PhD student at Northeastern University.

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

# A   PROOFS FOR SUM

The goal of this section is to prove Lemma 3.1 and Theorem 3.2.

## A.1   PROOFS FOR SCALAR SUM

We first state a technical lemma adapted from Lemma 4.12 in the work of Ghazi et al. (2021). Our lemma expresses divergence in terms of the per-instance distance between databases, which is necessary for the per-instance statement in Lemma 3.1.

**Lemma A.1.** *For any function $f : \mathcal{X}^n \to \mathbb{Z}$ and parameters $p \in (0, 1/2]$ and $m \in \mathbb{N}$, let $M_{m,p,f} : \mathcal{X}^n \to \mathbb{Z}$ be the algorithm that samples $\eta \sim \mathbf{Bin}(m, p)$ and returns $f(X) + \eta$. For any $t \in \mathbb{N}$, $X \sim X'$ such that $|f(X) - f(X')| \leq t$, and $\alpha \in (0, 1)$ where $\alpha m p \geq 2t$,*

$$D_\infty^\delta(M_{m,p,f}(X) \| M_{m,p,f}(X')) \leq \varepsilon$$

*where $\varepsilon = |f(X) - f(X')| \ln \frac{1+\alpha}{1-\alpha}$ and $\delta = 2 \exp(-\alpha^2 mp/10)$.*

*Proof.* Fix any set of integers $Z$. We will show that

$$\mathbb{P}\left[M_{m,p,f}(X) \in Z\right] \leq \exp\left(|f(X) - f(X')| \cdot \ln \frac{1+\alpha}{1-\alpha}\right) \mathbb{P}\left[M_{m,p,f}(X) \in Z\right]$$
$$+ 2\exp(-\alpha^2 mp/10)$$

which is equivalent to the divergence bound.

For any integer $i$, we use the notation $Z - i$ to denote the set $\{z - i \mid z \in Z\}$. Then

$$\mathbb{P}\left[M_{m,p,f}(X) \in Z\right] = \sum_{z \in Z} \mathbb{P}_{\eta \sim \mathbf{Bin}(m,p)}\left[\eta = z - f(X)\right]$$

$$= \sum_{\hat{z} \in Z - f(X)} \mathbb{P}_{\eta \sim \mathbf{Bin}(m,p)}\left[\eta = \hat{z}\right]$$

$$\leq \sum_{\hat{z} \in (Z - f(X)) \cap Q} \mathbb{P}_{\eta \sim \mathbf{Bin}(m,p)}\left[\eta = \hat{z}\right] + \sum_{\hat{z} \notin Q} \mathbb{P}_{\eta \sim \mathbf{Bin}(m,p)}\left[\eta = \hat{z}\right]. \quad (1)$$

The last inequality holds for any set $Q$. Our specific $Q$ is

$$Q := \left\{ q \in \mathbb{Z} \mid \frac{\mathbb{P}_{\eta \sim \mathbf{Bin}(m,p)}\left[\eta = q\right]}{\mathbb{P}_{\eta \sim \mathbf{Bin}(m,p)}\left[\eta = q + f(X) - f(X')\right]} < \exp\left(|f(X) - f(X')| \cdot \ln \frac{1+\alpha}{1-\alpha}\right) \right\}.$$

Recall that we supposed $|f(X) - f(X')| \leq t \leq \frac{\alpha m p}{2}$. The following fact comes from Lemma 4.12 in Ghazi et al. (2021): for any $k \in [-t, t]$,

$$\mathbb{P}_{Y \sim \mathbf{Bin}(m,p)}\left[\frac{\mathbb{P}_{\eta \sim \mathbf{Bin}(m,p)}\left[\eta = Y\right]}{\mathbb{P}_{\eta \sim \mathbf{Bin}(m,p)}\left[\eta = Y + k\right]} \geq \exp\left(|k| \cdot \ln \frac{1+\alpha}{1-\alpha}\right)\right] \leq 2\exp(-\alpha^2 mp/10).$$

Then by substitution,

$$(1) \leq \sum_{\hat{z} \in (Z - f(X)) \cap Q} \exp\left(|f(X) - f(X')| \cdot \ln \frac{1+\alpha}{1-\alpha}\right) \mathbb{P}_{\eta \sim \mathbf{Bin}(m,p)}\left[\eta = \hat{z} + f(X) - f(X')\right]$$

$$+ 2\exp(-\alpha^2 mp/10)$$

$$= \sum_{\hat{z} \in (Z - f(X')) \cap Q} \exp\left(|f(X) - f(X')| \cdot \ln \frac{1+\alpha}{1-\alpha}\right) \mathbb{P}_{\eta \sim \mathbf{Bin}(m,p)}\left[\eta = \hat{z}\right]$$

$$+ 2\exp(-\alpha^2 mp/10)$$

$$\leq \exp\left(|f(X) - f(X')| \cdot \ln \frac{1+\alpha}{1-\alpha}\right) \mathbb{P}\left[M_{m,p,f}(X') \in Z\right] + 2\exp(-\alpha^2 mp/10). \quad \square$$

With Lemma A.1 in hand, we now turn to proving our main result for scalar sum, Lemma 3.1.

**Lemma A.2** (Restatement of Lemma 3.1). *Fix any number of users $n$, $\varepsilon \leq 15$, and $0 < \delta < 1/2$. Let $g \geq \Delta\sqrt{n}$, $b > \frac{180g^2 \ln(2/\delta)}{\varepsilon^2 n}$, and $p = \frac{90g^2 \ln(2/\delta)}{b\varepsilon^2 n}$. Then 1) for any neighboring databases $X \sim X' \in [0, \Delta]^n$ that differ on user $u$, $D_\infty^\delta((\mathcal{S} \circ \mathcal{R}_{1D}^n)(X) \| (\mathcal{S} \circ \mathcal{R}_{1D}^n)(X')) \leq \varepsilon \cdot \left( \frac{2}{g} + \frac{|x_u - x_u'|}{\Delta} \right)$, and 2) for any input database $X \in [0, \Delta]^n$, the $\mathcal{P}_{1D}(X)$ is an unbiased estimate of $\sum_{i=1}^n x_i$ and has variance $O(\Delta + \frac{\Delta^2}{\varepsilon^2} \log \frac{1}{\delta})$.*

*Proof.* We first prove item 1. On input $X$, the shuffler produces a uniformly random permutation of the $(g + b)n$ bits $y_1, \ldots y_{(g+b)n}$. For each user $i$, let $z_i$ be the sum of the messages sent by user $i$, $z_i = \sum_{j=1}^{g+b} \vec{y}_{i,j}$. Then the random variable for permutations of $y_1, \ldots y_{(g+b)n}$ can also be viewed as a post-processing of the random variable $Z = \sum_{i=1}^n z_i$. Let $Z'$ be the analogue of $Z$ for input database $X'$, differing in user $u$. Then by the data-processing inequality, it suffices to prove $D_\infty^\delta(Z \| Z') \leq \varepsilon \cdot (2/g + |x_u - x_u'|/\Delta)$.

By the definition of $\mathcal{R}_{1D}$, $Z$ is distributed as $\hat{x}_u + \sum_{i \neq u} \hat{x}_u + \mathbf{Bin}(bn, p)$ while $Z'$ is distributed as $\hat{x}_u' + \sum_{i \neq u} \hat{x}_u + \mathbf{Bin}(bn, p)$. We have therefore reduced the analysis to that of the binomial mechanism. Our next step is to justify that Lemma A.1 applies to our construction. Specifically, we must identify $\alpha, m, t$ and show $p < 1/2$, $|\hat{x}_u - \hat{x}_u'| \leq t$, and $\alpha m p \geq 2t$.

We set $m = bn$, $t = g$, and $\alpha = \varepsilon/3g$. Our assumption on $b$ implies that $p < 1/2$. And the fact that $\hat{x}_i, \hat{x}_i'$ both lie in the interval $[0, g]$ implies $|\hat{x}_i - \hat{x}_i'| \leq g = t$. So it remains to prove $\alpha m p \geq 2t$:

$$
\begin{aligned}
\alpha m p &= \alpha \cdot bn \cdot \frac{90g^2 \ln(2/\delta)}{b\varepsilon^2 n} \\
&= \alpha \cdot \frac{90g^2 \ln(2/\delta)}{\varepsilon^2} \\
&= \frac{30g \ln(2/\delta)}{\varepsilon} \\
&> 2g \ln(2/\delta) && (\varepsilon < 15) \\
&> 2t. && (\delta < 1/2, t = g)
\end{aligned}
$$

Next, we derive the target bound on the divergence. Via Lemma A.1, we have that

$$
D_\infty^{\hat{\delta}}((\mathcal{S} \circ \mathcal{R}_{1D}^n)(X) \| (\mathcal{S} \circ \mathcal{R}_{1D}^n)(X')) \leq \hat{\varepsilon},
$$

where

$$
\begin{aligned}
\hat{\varepsilon} &= |\hat{x}_u - \hat{x}_u'| \ln \frac{1 + \alpha}{1 - \alpha} \\
&\leq |\hat{x}_u - \hat{x}_u'| \cdot 3\alpha && (\alpha \text{ small}) \\
&= \varepsilon \cdot \frac{1}{g} \cdot |\hat{x}_u - \hat{x}_u'| \\
&\leq \varepsilon \cdot \frac{1}{g} \cdot \left( 2 + \frac{|x_u - x_u'| \cdot g}{\Delta} \right) && (\text{Discretization}) \\
&= \varepsilon \cdot \left( \frac{2}{g} + \frac{|x_u - x_u'|}{\Delta} \right)
\end{aligned}
$$

and

$$
\begin{aligned}
\hat{\delta} &= 2 \exp(-\alpha^2 m p / 10) \\
&\leq 2 \exp\left( -\alpha^2 \cdot 9 \cdot \frac{g^2 \ln(2/\delta)}{\varepsilon^2} \right) = \delta.
\end{aligned}
$$

Moving to item 2, observe that the sum of the messages produced by user $i$ is $\overline{x}_i + \eta_1 + \eta_2$. Since $\eta_1$ was drawn from $\mathbf{Ber}(x_i g/\Delta - \overline{x}_i)$, $\mathbb{E}[\eta_1] = x_i g/\Delta - \overline{x}_i$ and $\text{Var}[\eta_1] \leq 1/4$ since any Bernoulli random variable has variance $\leq 1/4$. Meanwhile, $\eta_2$ is a sample from $\mathbf{Bin}(b, p)$ so $\mathbb{E}[\eta_2] = bp$ and $\text{Var}[\eta_2] = bp(1-p) \leq bp$. Recall our definition above of $z_i = \sum_{j=1}^{b+g} \vec{y}_{i,j}$. Then $\mathbb{E}[z_i] = x_i g/\Delta + bp$

and, by the independence of $\eta_1$ and $\eta_2$, $\text{Var}\left[z_i\right] \leq 1/4 + bp$. Extending this over all $n$ users yields $\mathbb{E}\left[\sum_{i=1}^{n} z_i\right] = \frac{g}{\Delta}\sum_{i=1}^{n} x_i + bnp$ and $\text{Var}\left[\sum_{i=1}^{n} z_i\right] \leq n/4 + bnp$. $\mathcal{A}_{1\text{D}}$ shifts and rescales this quantity, so its final output is unbiased

$$\mathbb{E}\left[\frac{\Delta}{g}\left(\left(\sum_{i=1}^{n} z_i\right) - bnp\right)\right] = \sum_{i=1}^{n} x_i$$

and, by our assumption that $g \geq \Delta\sqrt{n}$ and choice of $p$,

$$\text{Var}\left[\frac{\Delta}{g}\left(\left(\sum_{i=1}^{n} z_i\right) - bnp\right)\right] \leq \frac{\Delta^2}{g^2}\left(\frac{n}{4} + bnp\right) = O\left(1 + \frac{\Delta^2 bnp}{g^2}\right) = O\left(\frac{\Delta^2}{\varepsilon^2}\log\frac{1}{\delta}\right).$$

$\square$

### A.2 PROOFS FOR VECTOR SUM

Having completed our analysis of $\mathcal{P}_{1\text{D}}$, we now turn to $\mathcal{P}_{\text{VEC}}$. The first step in this direction is proving the following variant of advanced composition:

**Lemma A.3** (Restatement of Lemma 3.3). *Fix any $\gamma \in (0, 1)$ and neighboring databases $X, X' \in \mathcal{X}^n$. For all $j \in [d]$, suppose algorithm $\mathcal{A}_j : \mathcal{X}^n \to \mathcal{Y}$ use independent random bits and satisfies $D_\infty^{\delta_j}(\mathcal{A}_j(X)||\mathcal{A}_j(X')) \leq \varepsilon_j$. Let*

$$\varepsilon' = \sum_{j=1}^{d} \varepsilon_j(e^{\varepsilon_j} - 1) + 2\sqrt{\log(1/\gamma)\sum_{j=1}^{d}\varepsilon_j^2} \quad \text{and} \quad \delta' = \sum_{j=1}^{d}\delta_j + \gamma.$$

*Then $D_\infty^{\delta'}(\mathcal{A}_1(X), \ldots, \mathcal{A}_d(X)||\mathcal{A}_1(X')\ldots, \mathcal{A}_d(X')) \leq \varepsilon'$.*

Our proof follows the steps in (Dwork et al., 2010b; Asi et al., 2019) with some minor adaptations. We provide the detailed proof for completeness. We first provide some useful definitions.

**Definition A.4.** *Given random variables $Y$ and $Z$, the* KL Divergence *between $Y$ and $Z$ is*

$$D(Y||Z) = \mathbb{E}_{y \in Y}\left[\log\frac{\Pr[Y = y]}{\Pr[Z = z]}\right].$$

*The* Max Divergence *between $Y$ and $Z$ is*

$$D_\infty(Y||Z) = \max_{S \subseteq \text{supp}(Y)}\left[\log\frac{\Pr[Y \in S]}{\Pr[Z \in S]}\right].$$

*The* total variation distance *between $Y$ and $Z$ is*

$$\Delta(Y, Z) = \max_{S \subseteq \text{supp}(Y)}|P_Y(S) - P_Z(S)|.$$

The following lemma relating these notions comes from Dwork and Roth (2014).

**Lemma A.5** (Lemma 3.17 and 3.18 (Dwork and Roth, 2014)). *Given random variables $Y$ and $Z$,*

1. *We have both $D_\infty^\delta(Y||Z) \leq \varepsilon$ and $D_\infty^\delta(Z||Y) \leq \varepsilon$ if and only if there exists random variables $Y', Z'$ such that $\Delta(Y, Y') \leq \delta/(e^\varepsilon + 1)$ and $\Delta(Z, Z') \leq \delta/(e^\varepsilon + 1)$, and $D_\infty(Y'||Z') \leq \varepsilon$, and $D_\infty(Z'||Y') \leq \varepsilon$.*

2. *If $D_\infty(Y||Z) \leq \varepsilon$ and $D_\infty(Z||Y) \leq \varepsilon$, then $D(Y||Z) \leq \varepsilon(e^\varepsilon - 1)$.*

For the first claim, Dwork and Roth (2014) only explicitly proves that $D_\infty(Y'||Z') \leq \varepsilon$, but their construction of $Y'$ and $Z'$ also implies $D_\infty(Z'||Y') \leq \varepsilon$.

We now have the tools to prove our variant of advanced composition, Lemma A.3.

*Proof of Lemma A.3.* For any neighboring datasets $X$ and $X'$, let $Y = (Y_1, Y_2, \ldots, Y_d)$ and $Z = (Z_1, Z_2, \ldots, Z_d)$ denote the outcomes of running $\mathcal{A}_1, \ldots, \mathcal{A}_d$ on $X, X'$ respectively. By Fact 2.3 and the first part of Lemma A.5, there exist random variables $Y' = (Y_1', \ldots, Y_d')$ and $Z' = (Z_1', \ldots, Z_d')$ such that, for any $j \in [d]$, the following four inequalities hold: (i) $\Delta(Y_j, Y_j') \leq \delta_j/(e^{\varepsilon_j} + 1)$, (ii) $\Delta(Z_j, Z_j') \leq \delta_j/(e^{\varepsilon_j} + 1)$, (iii) $D_\infty(Y_j'||Z_j') \leq \varepsilon_j$, and (iv) $D_\infty(Z_j'||Y_j') \leq \varepsilon_j$. Then

$$\Delta(Y, Y'), \Delta(Z, Z') \leq \sum_{j=1}^d \frac{\delta_j}{e^{\varepsilon_j} + 1} < \frac{1}{2}\sum_{j=1}^d \delta_j.$$

Consider the outcome set $B = \{v \in \mathbb{R}^d : \Pr[Y' = v] \geq e^{\varepsilon'}\Pr[Z' = v]\}$. It remains to prove $\Pr[Y' \in B] \leq \gamma$. For any fixed realization $v = (v_1, \ldots, v_d) \in \mathbb{R}^d$, we have

$$\log\left(\frac{\Pr[Y' = v]}{\Pr[Z' = v]}\right) = \sum_{j=1}^d \log\left(\frac{\Pr[Y_j' = v_j]}{\Pr[Z_j' = v_j]}\right) \triangleq \sum_{j=1}^d c_j$$

where the first step follows from independence and the last step defines $c_j$. Since $D_\infty(Y_j'||Z_j') \leq \varepsilon_j$ and $D_\infty(Z_j'||Y_j') \leq \varepsilon_j$, we know that $|c_j| \leq \varepsilon_j$, and by the second part of Lemma A.5

$$\mathbb{E}_{v_j \sim Y_j}[c_j] = \mathbb{E}_{v_j \sim Y_j}\left[\log\left(\frac{\Pr[Y_j' = v_j]}{\Pr[Z_j' = v_j]}\right)\right] \leq \varepsilon_j(e^{\varepsilon_j} - 1).$$

Thus we have

$$\Pr[Y' \in B] = \Pr\left[\sum_{j=1}^d c_j \geq \varepsilon'\right]$$

$$= \Pr\left[\sum_{j=1}^d c_j \geq \sum_{j=1}^d \varepsilon_j(e^{\varepsilon_j} - 1) + 2\sqrt{\log(1/\gamma)\sum_{j=1}^d \varepsilon_j^2}\right] \leq \gamma.$$

by a Chernoff-Hoeffding bound. Tracing back, $\mathbb{P}[Y' \in B] \leq \gamma$ implies $D_\infty^\gamma(Y'||Z') \leq \varepsilon'$. Our previous bounds on $\Delta(Y, Y')$ and $\Delta(Z, Z')$ then give $D_\infty^{\sum_{j=1}^d \delta_j + \gamma}(Y||Z) \leq \varepsilon'$. $\square$

Finally, we can now prove Theorem 3.2, our primary guarantee for vector summation.

**Theorem A.6** (Restatement of Theorem 3.2). *For any $0 < \varepsilon \leq 15$, $0 < \delta < 1/2$, $d, n \in \mathbb{N}$, and $\Delta_2 > 0$, there are choices of parameters $b, g \in \mathbb{N}$ and $p \in (0, 1/2)$ for $\mathcal{P}_{1D}$ (Algorithm 3.1) such that, for inputs $\vec{X} = (\vec{x}_1, \ldots, \vec{x}_n)$ of vectors with maximum norm $||\vec{x}_i||_2 \leq \Delta_2$, 1) $\mathcal{P}_{VEC}$ is $(\varepsilon, \delta)$-shuffle private and, 2) $\mathcal{P}_{VEC}(\vec{X})$ is an unbiased estimate of $\sum_{i=1}^n \vec{x}_i$ and has bounded variance*

$$\mathbb{E}\left[\left\|\mathcal{P}_{VEC}(\vec{X}) - \sum_{i=1}^n \vec{x}_i\right\|_2^2\right] = O\left(\frac{d\Delta_2^2}{\varepsilon^2}\log^2\frac{d}{\delta}\right).$$

*Proof.* We set $g = \max(2\Delta_2\sqrt{n}, \sqrt{d}, 4)$. We will choose $b, p$ later in the proof.

Fix neighboring databases of vectors $\vec{X}$ and $\vec{X}'$. We assume without loss of generality that the databases differ only in the first user, $\vec{x}_1 \neq \vec{x}_1'$. Then the $\ell_2$ sensitivity is

$$\left\|\sum_{i=1}^n \vec{x}_i - \sum_{i=1}^n \vec{x}_i'\right\|_2 = \|\vec{x}_1 - \vec{x}_1'\|_2 \leq 2\Delta_2. \tag{2}$$

From our definition of $\vec{w}_{i,j}$ in line 4 of the pseudocode for $\mathcal{P}_{VEC}$ and the fact that $\|\vec{x}_i\|_2 \leq \Delta_2$, $\vec{w}_{i,j} \geq 0$ and $\vec{w}_{i,j} \leq \|\vec{x}_i\|_\infty + \Delta_2 \leq \|\vec{x}_i\|_2 + \Delta_2 \leq 2\Delta_2$. This means the sensitivity of the value of $\vec{w}_{i,j}$ is $2\Delta_2$.

For each coordinate $j \in [d]$, we use $a_j$ to denote the absolute difference in the $j$-th coordinate,

$$a_j := |\vec{x}_{1,j} - \vec{x}_{1,j}'| = |\vec{w}_{1,j} - \vec{w}_{1,j}'|$$

Let $\gamma, \hat{\delta} = \delta/(d+1)$ and $\hat{\varepsilon} = \varepsilon/18\sqrt{\log(1/\gamma)}$. Because $g \geq 2\Delta_2\sqrt{n}$, we may use Lemma 3.1 to set $b, p$ such that

$$D_\infty^{\hat{\delta}}((\mathcal{S} \circ \mathcal{R}_{1D}^n)(W_j) \| (\mathcal{S} \circ \mathcal{R}_{1D}^n)(W_j')) \leq \hat{\varepsilon} \cdot \left(\frac{2}{g} + \frac{a_j}{2\Delta_2}\right) \triangleq \varepsilon_j$$

where we use $W_j$ to denote $(w_{1,j}, \ldots, w_{n,j})$ and the last step defines $\varepsilon_j$. Notice that

$$\begin{aligned}
\sum_{j \in [d]} \varepsilon_j^2 &= \hat{\varepsilon}^2 \cdot \sum_{j \in [d]} \frac{4}{g^2} + \frac{2a_j}{\Delta_2 g} + \frac{a_j^2}{4\Delta_2^2} \\
&\leq \hat{\varepsilon}^2 \cdot \left(\frac{4d}{g^2} + \frac{2\|\vec{x}_1 - \vec{x}_1'\|_1}{\Delta_2 g} + \frac{\|\vec{x}_1 - \vec{x}_1'\|_2^2}{4\Delta_2^2}\right) && \text{(Def of } a_j) \\
&\leq \hat{\varepsilon}^2 \cdot \left(\frac{4d}{g^2} + \frac{2\sqrt{d}\|\vec{x}_1 - \vec{x}_1'\|_2}{\Delta_2 g} + \frac{\|\vec{x}_1 - \vec{x}_1'\|_2^2}{4\Delta_2^2}\right) \\
&\leq \hat{\varepsilon}^2 \cdot \left(\frac{4d}{g^2} + \frac{4\sqrt{d}}{g} + 1\right) && \text{(Via (2))} \\
&\leq \hat{\varepsilon}^2 \cdot (4 + 4 + 1) = 9\hat{\varepsilon}^2 && (3)
\end{aligned}$$

The final inequality comes from the fact that $g \geq \sqrt{d}$. By our choices of $\hat{\varepsilon}, \gamma$, and $\hat{\delta}$, we apply Lemma 3.3 to conclude that

$$\begin{aligned}
D_\infty^\delta((S \circ \mathcal{R}_{VEC}^n)(\vec{X}) \| (S \circ \mathcal{R}_{VEC}^n)(\vec{X}')) &\leq \sum_{j=1}^d \varepsilon_j(e^{\varepsilon_j} - 1) + 2\sqrt{\log(1/\gamma)\sum_{j=1}^d \varepsilon_j^2} \\
&\leq \sum_{j=1}^d 2\varepsilon_j^2 + 2\sqrt{\log(1/\gamma)\sum_{j=1}^d \varepsilon_j^2} \\
&\leq 18\hat{\varepsilon}^2 + 6\hat{\varepsilon}\sqrt{\log(1/\gamma)} && \text{(Via (3))} \\
&\leq \varepsilon && \text{(Choice of } \hat{\varepsilon})
\end{aligned}$$

which is precisely $(\varepsilon, \delta)$-differential privacy by Fact 2.3. Note that the second inequality holds because $g \geq 4$ implies $\varepsilon_j \leq \hat{\varepsilon} \leq \varepsilon/18$. Since $\varepsilon \leq 15$, we get $\varepsilon_j < 1$, and $e^{\varepsilon_j} - 1 \leq 2\varepsilon_j$.

The estimator produced by $\mathcal{P}_{VEC}$ is unbiased because each invocation of $\mathcal{P}_{1D}$ yields unbiased estimates. And we have, by linearity and the variance bound of $\mathcal{P}_{1D}$,

$$\mathbb{E}\left[\left\|\mathcal{P}_{VEC}(\vec{X}) - \sum_{i=1}^n \vec{x}_i\right\|_2^2\right] = \sum_{j=1}^d \mathbb{E}\left[\left|\mathcal{P}_{VEC}(\vec{X}_j) - \sum_{i=1}^n \vec{x}_{i,j}\right|^2\right] = O\left(\frac{d\Delta_2^2}{\hat{\varepsilon}^2}\log\left(\frac{1}{\hat{\delta}}\right)\right).$$

The theorem statement follows by substitution. $\square$

## A.3 ROBUST PRIVACY OF SUM

We now discuss the *robustness* of our privacy guarantees, as mentioned at the end of Section 3.1. In more detail, we slightly generalize the privacy analysis of $\mathcal{P}_{1D}$ to account for the case where a $\gamma$ fraction of the users are honest. Because $\mathcal{P}_{1D}$ is used as a building block for the other protocols we describe, this robustness carries over. We first present the definition of robust shuffle privacy originally given by Balcer et al. (2021).

**Definition A.7** (Balcer et al. (2021)). *Fix $\gamma \in (0, 1]$. A protocol $\mathcal{P} = (\mathcal{R}, \mathcal{A})$ is $(\varepsilon, \delta, \gamma)$-robustly shuffle differentially private if, for all $n \in \mathbb{N}$ and $\gamma' \geq \gamma$, the algorithm $\mathcal{S} \circ \mathcal{R}^{\gamma'n}$ is $(\varepsilon, \delta)$-differentially private. In other words, $\mathcal{P}$ guarantees $(\varepsilon, \delta)$-shuffle privacy whenever at least a $\gamma$ fraction of users follow the protocol.*

Although the above definition only explicitly handles drop-out attacks by $1 - \gamma$ malicious users, this is without loss of generality: any messages sent to the shuffler from the malicious users constitute a post-processing of $\mathcal{S} \circ \mathcal{R}^{\gamma n}$.

Now, we show that reducing the number of participants in $\mathcal{P}_{1D}$ loosens the bound on the divergence in a smooth fashion. Because $\mathcal{P}_{1D}$ is the building block of our other protocols, they inherit its robustness. The statement requires $\gamma \geq 1/3$, but this constant is arbitrary; changing it to a different value will only influence the lower bound on $g$.

**Claim A.8.** *Fix any $\gamma \in [1/3, 1]$, any number of users $n$, $\varepsilon \leq 1$, and $0 < \delta < 1$. If $g \geq 3$, $b > 180 \cdot \frac{g^2}{\varepsilon^2 n} \ln \frac{2}{\delta}$, and $p = 90 \cdot \frac{g^2}{b\varepsilon^2 n} \ln \frac{2}{\delta}$ then, for any neighboring databases $X \sim X' \in [0, \Delta]^n$ that differ on user $u$,*

$$D_\infty^\delta((S \circ \mathcal{R}_{1D}^{\gamma n})(X) \| (S \circ \mathcal{R}_{1D}^{\gamma n})(X')) \leq \frac{\varepsilon}{\gamma} \cdot \left( \frac{2}{g} + \frac{|x_u - x_u'|}{\Delta} \right).$$

*This implies $\mathcal{P}_{1D}$ satisfies $(O(\varepsilon/\gamma), \delta, \gamma)$-robust shuffle privacy.*

*Proof.* The proof proceeds almost identically with that of Item 2 in Lemma 3.1. The key change is that privacy noise is now drawn from $\mathbf{Bin}(\gamma bn, p)$. This means we take $m = \gamma bn$. We again set $t = g$ but now $\alpha = \varepsilon/3\gamma g$. Note that $\alpha < 1/3$ because $\varepsilon \leq 1$ and $\gamma g > 3\gamma > 1$.

As before, our assumption on $b$ implies that $p < 1/2$ and we have $|\hat{x}_u - \hat{x}_u'| \leq g = t$. So it remains to prove $\alpha mp \geq 2t$ in order to apply Lemma A.1:

$$\alpha mp = \alpha \cdot \gamma bn \cdot \frac{90 g^2 \ln(2/\delta)}{b\varepsilon^2 n} = \alpha \cdot \gamma \cdot \frac{90 g^2 \ln(2/\delta)}{\varepsilon^2} = \frac{30 g \ln(2/\delta)}{\varepsilon} > 2t$$

Now we derive the target bound on the divergence. Via Lemma A.1, we have that

$$D_\infty^{\hat{\delta}}((S \circ \mathcal{R}_{1D}^n)(X) \| (S \circ \mathcal{R}_{1D}^n)(X')) \leq \hat{\varepsilon},$$

where

$$\begin{aligned}
\hat{\varepsilon} &= |\hat{x}_u - \hat{x}_u'| \ln \frac{1 + \alpha}{1 - \alpha} \\
&\leq |\hat{x}_u - \hat{x}_u'| \cdot 3\alpha && (\alpha \text{ small}) \\
&= \frac{\varepsilon}{\gamma} \cdot \frac{1}{g} \cdot |\hat{x}_u - \hat{x}_u'| \\
&\leq \frac{\varepsilon}{\gamma} \cdot \frac{1}{g} \cdot \left( 2 + \frac{|x_u - x_u'| \cdot g}{\Delta} \right) && (\text{Discretization}) \\
&= \frac{\varepsilon}{\gamma} \cdot \left( \frac{2}{g} + \frac{|x_u - x_u'|}{\Delta} \right)
\end{aligned}$$

and

$$\begin{aligned}
\hat{\delta} &= 2 \exp(-\alpha^2 mp/10) \\
&= 2 \exp(-\alpha^2 \cdot 9\gamma \cdot \frac{g^2 \ln(2/\delta)}{\varepsilon^2}) \\
&= 2 \exp(-\frac{1}{\gamma} \cdot \ln(2/\delta)) \\
&\leq 2 \exp(-\ln(2/\delta)) = \delta
\end{aligned}$$

This concludes the proof. $\qquad\square$

When we replace $\varepsilon$ in the proof of Theorem 3.2 with $\varepsilon/\gamma$, we can invoke Claim A.8 to derive the following statement about the robust shuffle privacy of $\mathcal{P}_{VEC}$.

**Corollary A.9.** *For any $0 < \varepsilon \leq 1$, $0 < \delta < 1$, $d, n \in \mathbb{N}$, and $\Delta_2 > 0$, there are choices of parameters $b, g \in \mathbb{N}$ and $p \in (0, 1/2)$ for $\mathcal{P}_{1D}$ (Algorithm 3.1) such that, for inputs $\vec{X} = (\vec{x}_1, \ldots, \vec{x}_n)$ of vectors with maximum norm $\|\vec{x}_i\|_2 \leq \Delta_2$ and any $\gamma \in [1/3, 1]$, $\mathcal{P}_{VEC}$ is $(\varepsilon/\gamma, \delta, \gamma)$-robustly shuffle private*

Because $\mathcal{P}_{VEC}$ serves as a primitive for our optimization protocols, each round of those protocols ensures privacy in the face of two-thirds corruptions. For comparison, the generic construction by Beimel et al. (2020) is able to simulate arbitrary centrally private algorithms but requires an honest majority.

## B  PROOFS FOR SECTION 4

### B.1  PROOFS FOR SECTION 4.1

We start by proving the guarantee for $\mathcal{P}_{\text{SGD}}$, Theorem 4.1. The first step is to formally define the gradient oracle setup that will be necessary for several of our results.

**Definition B.1.** *Given function $\ell$ over $\Theta \subset \mathbb{R}^d$, we say $G \colon \Theta \to \mathbb{R}^d$ is an $L_G$-noisy gradient oracle for $\ell$ if for every $\theta \in \Theta$, (1) $\mathbb{E}_G\left[G(\theta)\right] \in \partial\ell(\theta)$, and (2) $\mathbb{E}_G\left[\|G(\theta)\|_2^2\right] \leq L_G^2$. Furthermore, we say the gradient oracle $G$ has* variance at most $\sigma^2$ *if $\mathbb{E}_G\left[\|G(\theta) - \mathbb{E}_G\left[G(\theta)\right]\|_2^2\right] \leq \sigma^2$.*

The following lemma is standard for (noisy) stochastic gradient descent.

**Lemma B.2** (Restatement of Lemma 4.2). *Suppose $\mathcal{P}_{\text{SGD}}$ queries an $L_G$-noisy gradient oracle at each iteration, then its output satisfies*

$$\mathbb{E}\left[\ell(\bar{\theta}, \mathcal{D})\right] \leq \min_{\theta \in \Theta} \ell(\theta, \mathcal{D}) + \frac{D^2}{2\eta T} + \frac{\eta L_G^2}{2}.$$

We now have the tools to prove Theorem 4.1.

*Proof of Theorem 4.1.* The privacy guarantee follows directly from the privacy guarantee of the vector summation protocol (see Theorem 3.2), and the fact that the algorithm queries each user at most once.

For the accuracy guarantee, our algorithm directly optimizes the excess population loss. For each time step $t \in \{0, 1, \ldots, T-1\}$, we write the gradient $\bar{g}_t = \frac{1}{b}\sum_{i=1}^{b}\nabla_\theta\ell(\theta_{t-1}, x_{t,i}) + \mathbf{Z}_t$. By Theorem 3.2, we know that

$$\mathbb{E}_{x_{t,1},\ldots,x_{t,b},\mathbf{Z}_t}\left[\bar{g}_t\right] = \mathbb{E}_{x\sim\mathcal{D}}\left[\nabla_\theta\ell(\theta_{t-1}, x)\right]$$

and

$$\mathbb{E}_{x_{t,1},\ldots,x_{t,b},\mathbf{Z}_t}\left[\|\bar{g}_t\|_2^2\right] = \mathbb{E}_{x_{t,1},\ldots,x_{t,b},\mathbf{Z}_t}\left[\left\|\frac{1}{b}\sum_{i=1}^{b}\nabla_\theta\ell(\theta_{t-1}, x_{t,i}) + \mathbf{Z}_t\right\|_2^2\right]$$

$$= \mathbb{E}_{x_{t,1},\ldots,x_{t,b}}\left[\left\|\frac{1}{b}\sum_{i=1}^{b}\nabla_\theta\ell(\theta_{t-1}, x_{t,i})\right\|_2^2\right] + \mathbb{E}\left[\|\mathbf{Z}_t\|_2^2\right]$$

$$\leq L^2 + O\left(\frac{dL^2\log^2(d/\delta)}{b^2\varepsilon^2}\right), \tag{4}$$

where the second equality uses the independence of the data and noise, and the inequality follows from $\ell$ being $L$-Lipschitz and Theorem 3.2.

The gradient $\bar{g}_t$ therefore qualifies as a noisy gradient oracle call with $L_G^2 = L^2 + O\left(\frac{dL^2\log^2(d/\delta)}{b^2\varepsilon^2}\right)$. By Lemma 4.2 and the fact that there are $n/b$ iterations, we get

$$\mathbb{E}\left[\ell(\bar{\theta}, \mathcal{D})\right] \leq \min_{\theta \in \Theta} \ell(\theta, \mathcal{D}) + \frac{D^2}{2\eta T} + \frac{\eta L_G^2}{2}$$

$$\leq \min_{\theta \in \Theta} \ell(\theta, \mathcal{D}) + \frac{DL_G}{\sqrt{T}} \qquad\qquad \text{(AM-GM ineq.)}$$

$$= \min_{\theta \in \Theta} \ell(\theta, \mathcal{D}) + D\sqrt{\frac{1}{T}\left(L^2 + O\left(\frac{dL^2\log^2(d/\delta)}{b^2\varepsilon^2}\right)\right)}$$

$$= \min_{\theta \in \Theta} \ell(\theta, \mathcal{D}) + O\left(DL\sqrt{\frac{b}{n} + \frac{d\log^2(d/\delta)}{\varepsilon^2 bn}}\right) \qquad\qquad (T = n/b)$$

$$= \min_{\theta \in \Theta} \ell(\theta, \mathcal{D}) + O\left(\frac{d^{1/4}DL\log^{1/2}(d/\delta)}{\sqrt{\varepsilon n}}\right),$$

The final step uses our choice of $b = \frac{\sqrt{d}\log(d/\delta)}{\varepsilon}$ $\qquad\square$

Next, we prove the guarantee for $\mathcal{P}_{\text{AGD}}$, Theorem 4.3.

*Proof of Theorem 4.3.* The privacy guarantee follows directly from the privacy guarantee of the vector summation protocol (see Theorem 3.2), and the fact that the algorithm queries each user at most once, and the remaining computation is post-processing.

We now prove the accuracy guarantee. For any iteration $t \in [T]$, let $\mathbf{Z}_t$ be the noise introduced by the summation protocol $\mathcal{P}_{\text{VEC}}$ and let $\bar{g}_t = \frac{1}{b}\sum_{i=1}^{b}\nabla_\theta\ell(\theta_{t-1}, x_{t,i}) + \mathbf{Z}_t$. We first bound the variance:

$$\mathbb{E}\left[\|G(\theta_{t-1}) - \mathbb{E}\left[G(\theta_{t-1})\right]\|_2^2\right]$$

$$= \mathbb{E}_{x_{t,1},\ldots,x_{t,b},\mathbf{Z}_t}\left[\left\|\frac{1}{b}\sum_{i=1}^{b}\nabla_\theta\ell(\theta_{t-1}, x_{t,i}) + \mathbf{Z}_t - \mathbb{E}_{x\sim\mathcal{D}}\left[\nabla\ell(\theta_{t-1}, x)\right]\right\|_2^2\right]$$

$$= \mathbb{E}_{x_{t,1},\ldots,x_{t,b}}\left[\left\|\frac{1}{b}\sum_{i=1}^{b}\nabla\ell(\theta_{t-1}, x_{t,i}) - \mathbb{E}_{x\sim\mathcal{D}}\left[\nabla\ell(\theta_{t-1}, x)\right]\right\|_2^2\right] + \mathbb{E}\left[\|\mathbf{Z_t}\|_2^2\right]$$

$$= \text{Var}\left[\frac{1}{b}\sum_{i=1}^{b}\nabla\ell(\theta_{t-1}, x_{t,i})\right] + \mathbb{E}\left[\|\mathbf{Z_t}\|_2^2\right]$$

$$= \frac{1}{b}\text{Var}\left[\nabla\ell(\theta_{t-1}, x_{t,1})\right] + \mathbb{E}\left[\|\mathbf{Z_t}\|_2^2\right]$$

$$= \frac{1}{b}\mathbb{E}_{x_t\sim\mathcal{D}}\left[\|\nabla\ell(\theta_{t-1}, x_t) - \mathbb{E}_x\left[\nabla\ell(\theta_{t-1}, x)\right]\|_2^2\right] + \mathbb{E}\left[\|\mathbf{Z_t}\|_2^2\right]$$

$$= O\left(\frac{L^2}{b} + \frac{dL^2\log^2(d/\delta)}{b^2\varepsilon^2}\right). \tag{5}$$

The second equality comes from the fact that the data and noise are independent, the fourth equality uses the independence of $x_{t,1},\cdots,x_{t,b}$ and (5) follows from Theorem 3.2. Next, we use the following result from Lan (2012).

**Lemma B.3** (Restatement of Lemma 4.4). *Suppose $\mathcal{P}_{\text{AGD}}$ receives a noisy gradient oracle with variance (at most) $\sigma^2$ in each iteration. Taking $L_t = \frac{1}{t+1}((T+2)^{3/2}\frac{\sigma}{D} + \beta), \alpha_t = \frac{2}{t+2}$, then the population loss of $\mathcal{P}_{\text{AGD}}$ can be bounded as*

$$\mathbb{E}\left[\ell(\bar{\theta}, \mathcal{D})\right] \leq \min_{\theta\in\Theta}\ell(\theta, \mathcal{D}) + O\left(\frac{\beta D^2}{T^2} + \frac{D\sigma}{\sqrt{T}}\right).$$

In our case, we have $\sigma^2 = \frac{L^2}{b} + \frac{dL^2\log^2(d/\delta)}{b^2\varepsilon^2}$ so that

$$\mathbb{E}\left[\ell(\bar{\theta}, \mathcal{D})\right] \leq \min_{\theta\in\Theta}\ell(\theta, \mathcal{D}) + O\left(\frac{\beta D^2}{T^2} + \frac{D\sigma}{\sqrt{T}}\right)$$

$$\leq \min_{\theta\in\Theta}\ell(\theta, \mathcal{D}) + O\left(\frac{\beta D^2}{T^2} + \frac{D\sqrt{\frac{L^2}{b} + \frac{dL^2\log^2(d/\delta)}{b^2\varepsilon^2}}}{\sqrt{T}}\right)$$

$$\leq \min_{\theta\in\Theta}\ell(\theta, \mathcal{D}) + O\left(\frac{\beta D^2}{T^2} + \frac{DL}{\sqrt{bT}} + \frac{\sqrt{d}DL\log(d/\delta)}{b\varepsilon\sqrt{T}}\right)$$

$$= \min_{\theta\in\Theta}\ell(\theta, \mathcal{D}) + O\left(\frac{b^2\beta D^2}{n^2} + \frac{DL}{\sqrt{n}} + \frac{\sqrt{d}DL\log(d/\delta)}{\varepsilon\sqrt{bn}}\right)$$

$$= \min_{\theta\in\Theta}\ell(\theta, \mathcal{D}) + O\left(\frac{d^{2/5}\beta^{1/5}D^{6/5}L^{4/5}\log^{4/5}(d/\delta)}{\varepsilon^{4/5}n^{4/5}} + \frac{DL}{\sqrt{n}}\right).$$

The first step follows from Lemma 4.4, and the second step comes from Eq. (5). In the last step, we substitute $b = \frac{n^{3/5}d^{1/5}L^{2/5}\log^{2/5}(d/\delta)}{\varepsilon^{2/5}\beta^{2/5}D^{2/5}}$. $\qquad\square$

Next, we prove our improved guarantee for convex losses by combining $\mathcal{P}_{\text{AGD}}$ and smoothing (Theorem 4.7).

*Proof of Theorem 4.7.* We directly optimize the loss function $\ell_\beta$ with $\mathcal{P}_{\text{AGD}}$. $\mathcal{P}_{\text{AGD}}$ only requires the gradient $\nabla\ell_\beta(\theta_t^{\text{md}}, x)$ ($t \in [T]$). By the second item in Lemma 4.6, this can be obtained by computing $\beta(\theta_t^{\text{md}} - \text{prox}_{\ell/\beta}(\theta_t^{\text{md}}))$. Moreover, since $\ell_\beta$ is $2L$-Lipschitz, $\mathcal{P}_{\text{AGD}}$ still guarantees $(\varepsilon, \delta)$-differential privacy with a constant scaling of the noise. For accuracy, we have that

$$
\mathbb{E}\left[\ell(\bar\theta, \mathcal{D})\right] - \min_{\theta\in\Theta}\ell(\theta, \mathcal{D}) \le \mathbb{E}\left[\ell_\beta(\bar\theta, \mathcal{D})\right] - \min_{\theta\in\Theta}\ell_\beta(\theta, \mathcal{D}) + \frac{L^2}{\beta}
$$

$$
\le O\left(\frac{DL}{\sqrt{n}}\right) + O\left(\frac{d^{2/5}\beta^{1/5}D^{6/5}L^{4/5}\log^{4/5}(d/\delta)}{\varepsilon^{4/5}n^{4/5}}\right) + \frac{L^2}{\beta}
$$

$$
\le O\left(\frac{DL}{\sqrt{n}} + \frac{d^{1/3}DL\log^{2/3}(d/\delta)}{\varepsilon^{2/3}n^{2/3}}\right).
$$

The first step follows from the third item in Lemma 4.6, the second step follows from the guarantee of $\mathcal{P}_{\text{AGD}}$ (Theorem 4.3) and the fact that $\ell_\beta$ is $\beta$-smooth, and the last step comes from our choice of $\beta = \frac{\varepsilon^{2/3}n^{2/3}L}{d^{1/3}D\log^{2/3}(d/\delta)}$. □

Finally, we prove Theorem 4.8, which applies to strongly convex $\ell$ and is the final result about sequentially interactive protocols.

*Proof of Theorem 4.8.* The privacy guarantee follows from the previously-proven privacy guarantee for $\mathcal{P}_{\text{AGD}}$ and the fact that we split the users into disjoint subsets. The rest of the proof focuses on the population loss.

We begin with the non-smooth setting. The high level idea for this case is that both the population loss and the distance to optimal solution shrink in each iteration. By recursively applying the shuffle private convex algorithm and solving the recursion, one can get improved convergence rate. Let $\bar\theta_i$ be the output of the $i$-th phase and $\theta^* = \arg\min_{\theta\in\Theta}\ell(\theta, \mathcal{D})$. Denote $D_i^2 = \mathbb{E}\left[\|\bar\theta_i - \theta^*\|_2^2\right]$ and $\Delta_i = \mathbb{E}\left[\ell(\bar\theta_i, \mathcal{D}) - \ell(\theta^\star, \mathcal{D})\right]$.

Due to the $\lambda$-strong convexity, we have $D_i^2 \le \frac{2\Delta_i}{\lambda}$. Since the guarantee of Theorem 4.7 uses $D$ as a bound on the distance between the starting point and optimum, we have

$$
\Delta_{i+1} \le C\left(\frac{D_iL}{\sqrt{n_i}} + \frac{d^{1/3}D_iL\log^{2/3}(d/\delta)}{\varepsilon^{2/3}n_i^{2/3}}\right) \le \sqrt{\frac{2\Delta_i}{\lambda}}\cdot C\left(\frac{L}{\sqrt{n_i}} + \frac{d^{1/3}L\log^{2/3}(d/\delta)}{\varepsilon^{2/3}n_i^{2/3}}\right)
$$
$$
:= \sqrt{\Delta_i}E
$$

Here $C$ is an universal constant, and the last step defines $E$. Hence, one has

$$
\frac{\Delta_{i+1}}{E^2} \le \sqrt{\frac{\Delta_i}{E^2}}.
$$

We know that $\Delta_1 \le 2L^2/\lambda$ (due to strong convexity), $E^2 \ge 2L^2/\lambda n$ (due to definition), and therefore, $\Delta_1/E^2 \le O(n)$. Hence, after $k = O(\log\log n)$ phases, we have $\Delta_k/E^2 \le 2$, and the population loss is

$$
\mathbb{E}\left[\ell(\bar\theta, \mathcal{D})\right] \le \min_{\theta\in\Theta}\ell(\theta, \mathcal{D}) + O\left(\frac{1}{\lambda}\cdot\left(\frac{L}{\sqrt{n_k}} + \frac{d^{1/3}L\log^{2/3}(d/\delta)}{n_k^{2/3}\varepsilon^{2/3}}\right)^2\right)
$$

$$
\le \min_{\theta\in\Theta}\ell(\theta, \mathcal{D}) + \tilde{O}\left(\frac{L^2}{\lambda n} + \frac{d^{2/3}L^2\log^{4/3}(d/\delta)}{\lambda\varepsilon^{4/3}n^{4/3}}\right).
$$

This completes the proof for the non-smooth setting. Turning to the setting where $\ell$ is also $\beta$-smooth, let $A = \frac{d^{2/5}\beta^{1/5}L^{4/5}\log^{4/5}(d/\delta)}{\varepsilon^{4/5}}$. Then our algorithm guarantees

$$\Delta_{i+1} \leq C \left( \frac{AD_i^{6/5}}{n_i^{4/5}} + \frac{LD_i}{\sqrt{n_i}} \right) \leq 2C \left( \frac{A\Delta_i^{3/5}}{\lambda^{3/5}n_i^{4/5}} + \frac{L\sqrt{\Delta_i}}{\sqrt{\lambda n_i}} \right) \tag{6}$$

for some constant $C > 1$.

First, suppose there exists $i \in [k]$ such that $\Delta_i$ is already small, i.e.

$$\Delta_i \leq 2^{10}C^4 \left( \frac{L^2}{\lambda n_i} + \frac{A^{5/2}}{\lambda^{3/2}n_i^2} \right) := T.$$

Then we claim $\Delta_t \leq 2^{10}C^4 \left( \frac{L^2}{\lambda n_i} + \frac{A^{5/2}}{\lambda^{3/2}n_i^2} \right)$ holds for all $t \geq i$. We divide into cases.

Case 1. Suppose $\frac{L^2}{\lambda n_i} \leq \frac{A^{5/2}}{\lambda^{3/2}n_i^2}$. Then we have

$$\begin{aligned}
\Delta_{i+1} &\leq 4C \cdot \left( 2^6 C^3 \frac{A^{5/2}}{\lambda^{2/3}n_{i+1}^2} + 2^5 C^3 \frac{LA^{5/4}}{\lambda^{5/4}n_{i+1}^{3/2}} \right) \\
&\leq 4C \cdot \left( 2^6 C^3 \frac{A^{5/2}}{\lambda^{2/3}n_i^2} + 2^5 C^3 \frac{A^{5/2}}{\lambda^{2/3}n_i^2} \right) \\
&\leq 2^{10}C^4 \frac{A^{5/2}}{\lambda^{3/2}n_i^2}.
\end{aligned}$$

The first step follows from Eq. (6), $n_i = n_{i+1} = n/k$, the second step follows our assumption.

Case 2. Suppose $\frac{L^2}{\lambda n_i} \geq \frac{A^{5/2}}{\lambda^{3/2}n_i^2}$. Then we have

$$\begin{aligned}
\Delta_{i+1} &\leq 4C \cdot \left( 2^6 C^3 \frac{AL^{6/5}}{\lambda^{6/5}n_{i+1}^{7/5}} + 2^5 C^3 \frac{L^2}{\lambda n_{i+1}} \right) \\
&\leq 4C \cdot \left( 2^6 C^3 \frac{L^2}{\lambda n_i} + 2^5 C^3 \frac{L^2}{\lambda n_i} \right) \\
&\leq 2^{10}C^4 \frac{L^2}{\lambda n_i}.
\end{aligned}$$

The first step follows from Eq. (6), $n_i = n_{i+1} = n/k$, the second step follows our assumption.

Next, suppose $\Delta_i \leq T = 2^{10}C^4 \left( \frac{L^2}{\lambda n_i} + \frac{A^{5/2}}{\lambda^{3/2}n_i^2} \right)$, then we claim that $\Delta_{i+1}$ is decreasing, i.e., $\Delta_{i+1} \leq \Delta_i$. Suppose this does not hold, since we have one of the following should hold

$$2C \frac{A\Delta_i^{3/5}}{\lambda^{3/5}n_{i+1}^{4/5}} \geq \frac{\Delta_i}{2} \quad \text{or} \quad 2C \frac{L\sqrt{\Delta_i}}{\sqrt{\lambda n_{i+1}}} \geq \frac{\Delta_i}{2}.$$

We argue this can not happens, since the former one implies $\Delta_i \leq 2^5 C^{5/2} \frac{A^{5/2}}{\lambda^{3/2}n_i^2}$ and the later implies $\Delta_i \leq 2^4 C^2 \frac{L}{\lambda n_i}$.

Combining the above argument, we know that $\Delta_i$ is decreasing, until it goes below a threshold $T$, and after that, the value $\Delta_i$ will always below such the threshold $T$. Now, suppose after $k = O(\log\log n)$ iterations, $\Delta_k$ is still above the threshold $T$, then we know $\{\Delta_i\}_{i \in [k]}$ is decreasing and we divide the algorithmic procedure into two phases. In the first phase, we have $\frac{L\sqrt{\Delta_i}}{\sqrt{\lambda n}} \leq \frac{A\Delta_i^{3/5}}{\lambda^{3/5}n_i^{4/5}}$, this happens when $\Delta_i \geq \frac{L^{10}\lambda}{A^{10}n_i^3}$, and the algorithm satisfies

$$\Delta_{i+1} \leq 4C \frac{A\Delta_i^{3/5}}{\lambda^{3/5}n_{i+1}^{4/5}} = 4C \frac{A\Delta_i^{3/5}}{\lambda^{3/5}n_i^{4/5}}.$$

For the second phase, we have $\frac{L\sqrt{\Delta_i}}{\sqrt{\lambda}n} \geq \frac{A\Delta_i^{3/5}}{\lambda^{3/5}n^{4/5}}$ and this happens when this happens when $\Delta_i \leq \frac{L^{10}\lambda}{A^{10}n_i^3}$, and the algorithm satisfies

$$\Delta_{i+1} \leq 4C\frac{L\sqrt{\Delta_i}}{\sqrt{\lambda}n_{i+1}} = 4C\frac{L\sqrt{\Delta_i}}{\sqrt{\lambda}n_i}.$$

We know one of the two phases will run for more than $\Omega(\log\log n)$ iterations. Suppose the former case holds, i.e., $\Delta_{i+1} \leq 4C\frac{A\Delta_i^{3/5}}{\lambda^{3/5}n_i^{4/5}}$. Let $E = \left(\frac{4CA}{\lambda^{3/5}n_i^{4/5}}\right)^{5/2} \leq 2^5C^{5/2}\frac{A^{5/2}}{\lambda^{3/2}n_i^2}$. Then, we have

$$\frac{\Delta_{i+1}}{E} \leq \left(\frac{\Delta_i}{E}\right)^{3/5}.$$

It is easy to verify that $\Delta_1/E \leq \text{poly}(n)$, and therefore, after $\Omega(\log\log n)$ iterations, we know that $\Delta_i$ will drop below $O(E) = 2^5C^{5/2}\frac{A^{5/2}}{\lambda^{3/2}n_i^2} \leq T$.

When later case holds, i.e., $\Delta_{i+1} \leq 4C\frac{L\sqrt{\Delta_i}}{\sqrt{\lambda}n_i}$. Let $E = 16C^2\frac{L^2}{\lambda n_i}$, then we know that

$$\frac{\Delta_{i+1}}{E} \leq \left(\frac{\Delta_i}{E}\right)^{1/2}.$$

It is easy to verify that $\Delta_1/E \leq \text{poly}(n)$, and therefore, after $\Omega(\log\log n)$ iterations, we know that $\Delta_i$ will drop below $O(E) = 16C^2\frac{L^2}{\lambda n_i} \leq T$.

In summary, taking $k = \Omega(\log\log n)$, we know that

$$\mathbb{E}\left[\ell(\bar{\theta}_k, \mathcal{D})\right] - \min_{\theta \in \Theta}\ell(\theta, \mathcal{D}) = \Delta_k \leq T = 2^{10}C^4\left(\frac{L^2}{\lambda n_k} + \frac{A^{5/2}}{\lambda^{3/2}n_k^2}\right)$$
$$\leq \tilde{O}\left(\frac{L^2}{\lambda n} + \frac{d\beta^{1/2}L^2\log(d/\delta)^2}{\lambda^{3/2}\varepsilon^2 n^2}\right).$$

This concludes the proof.

$\square$

## B.2 PROOFS FOR SECTION 4.2

We now turn to proofs for our fully interactive protocol, $\mathcal{P}_{\text{GD}}$. The first step will be proving Lemma B.6, which provides a privacy and population loss guarantee for $\mathcal{P}_{\text{GD}}$ when the loss function $\ell$ is smooth. We will later combine this analysis for smooth losses with the smoothing trick used in the previous section to obtain guarantees for non-smooth losses. Since $\mathcal{P}_{\text{GD}}$ passes the training data more than once and does not directly optimize the excess population loss, the proof of Lemma B.6 proceeds by bounding the empirical loss and the generalization error separately. First, we recall a result bounding empirical loss.

**Lemma B.4** (Bubeck et al. (2015)). *Let $\ell$ be convex and L-Lipschitz over closed convex set $\Theta \subset \mathbb{R}^d$ of diameter D. Let $\sigma^2$ denote the variance of the privacy noise in the gradient update step of $\mathcal{P}_{\text{GD}}$. For any set of data points S and $\eta > 0$, the output of $\mathcal{P}_{\text{GD}}$ satisfies*

$$\mathbb{E}\left[\ell(\bar{\theta}, S)\right] - \min_{\theta \in \Theta}\ell(\theta, S) \leq \frac{D^2}{2\eta T} + \frac{\eta(L^2 + \sigma^2)}{2}.$$

Next, we bound generalization error.

**Lemma B.5** (Lemma 3.4 (Bassily et al., 2019)). *Suppose $\ell$ is convex, L-Lipschitz, and $\beta$-smooth over closed convex set $\Theta \subset \mathbb{R}^d$ of diameter D and $\eta \leq \frac{2}{\beta}$. Then $\mathcal{P}_{\text{GD}}$ is $\alpha$-uniformly stable with $\alpha = L^2\frac{T\eta}{n}$. In other words, it satisfies*

$$\mathbb{E}_{S \sim \mathcal{D}^n}\left[|\ell(\bar{\theta}, S) - \ell(\bar{\theta}, \mathcal{D})|\right] \leq L^2\frac{T\eta}{n}.$$

With these two tools, we can state and prove Lemma B.6

**Lemma B.6.** *Let $\ell$ be convex, $L$-Lipschitz, and $\beta$-smooth over a closed convex set $\Theta \subset \mathbb{R}^d$ of diameter $D$. Then $\mathcal{P}_{\mathrm{GD}}$ is $(\varepsilon, \delta)$-shuffle private with population loss*

$$\mathbb{E}\left[\ell(\bar{\theta}, \mathcal{D})\right] \leq \min_{\theta \in \Theta} \ell(\theta, \mathcal{D}) + O\left(\frac{DL}{\sqrt{n}} + \frac{d^{1/2}DL \log^{3/2}(d/\delta)}{\varepsilon n}\right).$$

*Proof.* For the privacy guarantee, since each iteration guarantees $(\frac{\varepsilon}{2\sqrt{2T\log(1/\delta)}}, \frac{\delta}{T+1})$-differential privacy, and the algorithm runs for $T$ iterations in total, advanced composition implies $(\varepsilon, \delta)$-shuffle privacy overall.

We now focus on the loss guarantee. First, by Lemma B.4, the empirical loss satisfies

$$\mathbb{E}\left[\ell(\bar{\theta}; S)\right] \leq \min_{\theta \in \Theta} \ell(\theta; S) + \frac{D^2}{2\eta T} + \frac{\eta L_G^2}{2}$$

$$\leq \min_{\theta \in \Theta} \ell(\theta; S) + \frac{D^2}{2\eta T} + \frac{\eta L^2}{2} + O\left(\frac{\eta dTL^2 \log^3(nd/\delta)}{2n^2\varepsilon^2}\right)$$

where the second step follows from $L_G^2 \leq L^2 + O(\frac{dTL^2 \log^3(nd/\delta)}{n^2\varepsilon^2})$ (see Eq. (4)). Moreover, by Lemma B.5, we know the generalization error is most $L^2 \frac{T\eta}{n}$. Hence, for $S \sim \mathcal{D}^n$,

$$\mathbb{E}\left[\ell(\bar{\theta}; \mathcal{D})\right] - \min_{\theta \in \Theta} \ell(\theta, \mathcal{D}) \leq \mathbb{E}\left[|\ell(\bar{\theta}, \mathcal{D}) - \ell(\bar{\theta}, S)|\right] + \mathbb{E}\left[\ell(\bar{\theta}, S) - \min_{\theta \in \Theta} \ell(\theta, S)\right]$$

$$\leq \frac{D^2}{2\eta T} + \frac{\eta L^2}{2} + \frac{\eta dTL^2 \log^3(nd/\delta)}{2n^2\varepsilon^2} + L^2 \frac{T\eta}{n}$$

$$\leq O\left(\frac{DL\sqrt{d\log^3(nd/\delta)}}{\varepsilon n} + \frac{DL}{\sqrt{n}}\right)$$

The second step follows from Lemma B.5 and Lemma 4.2. Especially, we take expectation over $S \sim \mathcal{D}^n$ in Lemma 4.2. The last step follows by taking $\eta = \frac{D}{L\sqrt{T}}$ and $T = \min\{n, \frac{\varepsilon^2 n^2}{d\log^2(nd/\delta)}\}$. We use $\delta \ll 1/n$ for the final statement. $\qquad\square$

We can now prove Theorem 4.9. As done previously, we use Moreau envelope smoothing to apply the guarantee of Lemma B.6. For the strongly convex case, we apply the same reduction to the convex case as used by Feldman et al. (2020).

*Proof of Theorem 4.9.* The privacy guarantees are inherited from Lemma B.6. We employ Moreau envelope smoothing, as described in Lemma 4.6 and used earlier in the proof of Theorem 4.7, to construct $\ell_\beta$ with $\beta = \frac{L}{D} \min\left(\sqrt{n}, \frac{\varepsilon n}{d^{1/2}\log^{3/2}(nd/\delta)}\right)$. Optimizing $\ell_\beta$ yields

$$\mathbb{E}_{S \sim \mathcal{D}^n}\left[\ell(\bar{\theta}; \mathcal{D})\right] - \min_{\theta \in \Theta} \ell(\bar{\theta}, \mathcal{D})$$

$$\leq \mathbb{E}_{S \sim \mathcal{D}^n}\left[\ell_\beta(\bar{\theta}; \mathcal{D})\right] - \min_{\theta \in \Theta} \ell_\beta(\bar{\theta}, \mathcal{D}) + \frac{L^2}{2\beta}$$

$$\leq O\left(\frac{DL}{\sqrt{n}} + \frac{d^{1/2}DL \log^{3/2}(d/\delta)}{\varepsilon n}\right) + LD \cdot \left(\frac{1}{2\sqrt{n}} + \frac{d^{1/2}\log^{3/2}(d/\delta)}{2\varepsilon n}\right)$$

$$= O\left(\frac{DL}{\sqrt{n}} + \frac{d^{1/2}DL \log^{3/2}(d/\delta)}{\varepsilon n}\right).$$

The first step follows from the third item in Lemma 4.6, and the second step follows from Lemma B.6 and the definition of $\beta$.

For strongly convex losses, we use the same reduction as (Feldman et al., 2020).

**Lemma B.7** (Theorem 5.1 in (Feldman et al., 2020)). *Suppose for any loss function that is convex and $L$-Lipschitz over a closed convex set $\Theta \subset \mathbb{R}^d$ of diameter $D$, there is an $(\varepsilon, \delta)$-differentially private algorithm that achieves excess population loss of order*

$$LD\left(\frac{1}{\sqrt{n}} + \frac{\sqrt{d}}{\rho n}\right).$$

*Then for any $\lambda$-strongly convex, $L$-Lipschitz loss function, there is an $(\varepsilon, \delta)$-differentially private algorithm that achieves excess population loss of order*

$$\frac{L^2}{\lambda}\left(\frac{1}{n} + \frac{d}{\rho^2 n^2}\right).$$

We note that this reduction works for the fully interactive model and keeps the privacy guarantee. In particular, one can divide users into the same set of groups as (Feldman et al., 2020) and apply $\mathcal{P}_{\mathrm{GD}}$ sequentially to these groups. By taking $\rho = \frac{\varepsilon}{\log^{3/2}(nd/\delta)}$, we get the desired guarantee for strongly convex functions. $\qquad\square$

### B.3 SCO COMMUNICATION AND RUNTIME

Recall from the discussion at the end of Section 3.1 that, with the scaling by $\Delta$ trick, each user sends $O(d[\sqrt{n} + \log(1/\delta)/\varepsilon^2])$ bits in each $n$-user invocation of $\mathcal{P}_{\mathrm{VEC}}$, and the analyzer processes these in time $O(dn[\sqrt{n} + \log(1/\delta)/\varepsilon^2])$. Communication and runtime guarantees each for SCO algorithm then follow as simple corollaries based on the number of rounds and the batch size in each round.

**Corollary B.8.** *In the algorithm described in Theorem 4.1, each user sends $O(d[\sqrt{b} + \log(1/\delta)/\varepsilon^2])$ bits and the total runtime is*

$$O\left(dn\left[\sqrt{b} + \frac{\log(1/\delta)}{\varepsilon^2}\right]\right) = O\left(dn\left[d^{1/4}\sqrt{\frac{\log(d/\delta)}{\varepsilon}} + \frac{\log(1/\delta)}{\varepsilon^2}\right]\right).$$

**Corollary B.9.** *In the algorithm described in Theorem 4.3, each user sends $O(d[\sqrt{b} + \log(1/\delta)/\varepsilon^2])$ bits and the total runtime is*

$$O\left(dn\left[\sqrt{b} + \frac{\log(1/\delta)}{\varepsilon^2}\right]\right) = O\left(dn\left[\frac{d^{1/10}n^{3/10}L^{1/5}\log^{1/5}(d/\delta)}{\varepsilon^{1/5}\beta^{1/5}D^{1/5}} + \frac{\log(1/\delta)}{\varepsilon^2}\right]\right).$$

**Corollary B.10.** *In the algorithm described in Theorem 4.7, each user sends $O(d[\sqrt{b} + \log(1/\delta)/\varepsilon^2])$ bits and the total runtime is*

$$O\left(nd\left[\sqrt{b} + \frac{\log(1/\delta)}{\varepsilon^2}\right]\right) = O\left(nd\left[\frac{d^{1/6}n^{1/6}\log^{1/3}(d/\delta)}{\varepsilon^{1/3}} + \frac{\log(1/\delta)}{\varepsilon^2}\right]\right).$$

Note that the preceding corollary omits the time complexity of the smoothing step. A short discussion of doing this efficiently appears in Section 4 of the work of Bassily et al. (2019).

**Corollary B.11.** *In the algorithm described in Theorem 4.9, each user sends*

$$O\left(dT\left[\sqrt{n} + \frac{\log(1/\delta)}{\varepsilon^2}\right]\right) = \min\left\{n, \frac{\varepsilon^2 n^2}{d\log^2(nd/\delta)}\right\} \cdot O\left(d\left[\sqrt{n} + \frac{\log(1/\delta)}{\varepsilon^2}\right]\right)$$

*bits in total, and the total runtime is*

$$O\left(dnT\left[\sqrt{n} + \frac{\log(1/\delta)}{\varepsilon^2}\right]\right) = \min\left\{n, \frac{\varepsilon^2 n^2}{d\log^2(nd/\delta)}\right\} \cdot O\left(dn\left[\sqrt{n} + \frac{\log(1/\delta)}{\varepsilon^2}\right]\right).$$

## C PAN PRIVATE CONVEX OPTIMIZATION

Finally, we turn to pan-privacy, where we assume users/data arrive in a stream, and the goal is to guarantee differential privacy against a single intrusion into the stream-processing algorithm's internal state. We formalize the model and provide definition in Section C.1, and provide algorithms in Section C.2. Pan-privacy was introduced by Dwork et al. (2010a); we use the presentation given by Amin et al. (2020) and Balcer et al. (2021).

## C.1 MODELS AND DEFINITION

Our focus will be on online algorithms. They receive raw data one element at a time in a stream. At each step in the stream, the algorithm receives a data point, updates its internal state based on this data point, and then proceeds to the next element. The only way the algorithm "remembers" past elements is through its internal state. The formal definition we use is below.

**Definition C.1.** *An* online algorithm $\mathcal{Q}$ *is defined by an internal algorithm $\mathcal{Q}_{\mathcal{I}}$ and an output algorithm $\mathcal{Q}_{\mathcal{O}}$. $\mathcal{Q}$ processes a stream of elements through repeated application of $\mathcal{Q}_{\mathcal{I}} : \mathcal{X} \times \mathcal{I} \to \mathcal{I}$, which (with randomness) maps a stream element and internal state to an internal state. At the end of the stream, $\mathcal{Q}$ publishes a final output by executing $\mathcal{Q}_{\mathcal{O}}$ on its final internal state.*

As in the case of datasets, we say that two streams $X$ and $X'$ are *neighbors* if they differ in at most one element. Pan-privacy requires the algorithm's internal state and output to be differentially private with regard to neighboring streams.

**Definition C.2** (Dwork et al. (2010a); Amin et al. (2020)). *Given an online algorithm $\mathcal{Q}$, let $\mathcal{Q}_{\mathcal{I}}(X)$ denote its internal state after processing stream $X$, and let $X_{\leq t}$ be the first $t$ elements of $X$. We say $\mathcal{Q}$ is $(\varepsilon, \delta)$-pan-private if, for every pair of neighboring streams $X$ and $X'$, every time $t$ and every set of internal state, output state pairs $T \subset \mathcal{I} \times \mathcal{O}$,*

$$\mathbb{P}_{\mathcal{Q}}\left[\left(\mathcal{Q}_{\mathcal{I}}(X_{\leq t}), \mathcal{Q}_{\mathcal{O}}(\mathcal{Q}_{\mathcal{I}}(X))\right) \in T\right] \leq e^{\varepsilon} \cdot \mathbb{P}_{\mathcal{Q}}\left[\left(\mathcal{Q}_{\mathcal{I}}(X'_{\leq t}), \mathcal{Q}_{\mathcal{O}}(\mathcal{Q}_{\mathcal{I}}(X'))\right) \in T\right] + \delta. \quad (7)$$

*When $\delta = 0$, we say $\mathcal{Q}$ is $\varepsilon$-pan-private.*

## C.2 ALGORITHMS

Due to the structural similarity between sequential shuffle privacy and pan-privacy, all the results developed in Section 4.1 have pan-private counterparts.

**Theorem C.3.** *Suppose a convex loss function $\ell$ is L-Lipschitz over a closed convex set $\Theta \subset \mathbb{R}^d$, then there exists an algorithm with output $\bar{\theta}$ that guarantees $(\varepsilon, \delta)$-pan privacy, and:*

*(1) When the loss function is non-smooth, then the population loss satisfies*

$$\ell(\bar{\theta}, \mathcal{D}) \leq \min_{\theta \in \Theta} \ell(\theta, \mathcal{D}) + O\left(\frac{DL}{\sqrt{n}} + \frac{d^{1/3} DL \log^{1/3}(1/\delta)}{\varepsilon^{2/3} n^{2/3}}\right).$$

*(2) When the loss function is $\beta$-smooth, then the population loss satisfies*

$$\ell(\bar{\theta}, \mathcal{D}) \leq \min_{\theta \in \Theta} \ell(\theta, \mathcal{D}) + O\left(\frac{DL}{\sqrt{n}} + \frac{d^{2/5} \beta^{1/5} D^{6/5} L^{4/5} \log^{2/5}(1/\delta)}{\varepsilon^{4/5} n^{4/5}}\right).$$

*(3) When the loss function is $\lambda$-strongly convex, then the population loss satisfies*

$$\ell(\bar{\theta}, \mathcal{D}) \leq \min_{\theta \in \Theta} \ell(\theta, \mathcal{D}) + \tilde{O}\left(\frac{L^2}{\lambda n} + \frac{d^{2/3} L^2 \log^{2/3}(1/\delta)}{\lambda \varepsilon^{4/3} n^{4/3}}\right).$$

*(4) When the loss function is $\lambda$-strongly convex and $\beta$-smooth then the the population loss satisfies*

$$\ell(\bar{\theta}, \mathcal{D}) \leq \min_{\theta \in \Theta} \ell(\theta, \mathcal{D}) + \tilde{O}\left(\frac{L^2}{\lambda n} + \frac{d\beta^{1/2} L^2 \log(1/\delta)}{\lambda^{3/2} \varepsilon^2 n^2}\right).$$

*Proof.* We analyze the pseudocode presented in Algorithm 6. Although it does not explicitly obey the syntax of an online algorithm given in Definition C.1, it is straightforward to see that the internal state is the tuple $(\theta_t, \theta_t^{\mathsf{ag}}, \theta_t^{\mathsf{md}}, \bar{g}_t)$ and an update to $\bar{g}_t$ occurs when any data point is read. The updates to the $\theta$ variables also occur after the last data point of a batch is read.

We first prove Algorithm 6 is $(\varepsilon, \delta)$-pan private.

**Lemma C.4.** *For $0 < \varepsilon, \delta < 1$, Algorithm 6 is $(\varepsilon, \delta)$-pan-private.*

---

**Algorithm 6** Pan-private AC-SA

---

**Require:** Batch size $b$, privacy parameter $\varepsilon$, stream length $n$, learning rate sequence $\{L_t\}, \{\alpha_t\}$
1: Initialize noise parameter $\zeta^2 \leftarrow \frac{8L^2 \log(2/\delta)}{b^2 \varepsilon^2}$
2: Initialize parameter estimate $\theta_1^{\mathsf{ag}} = \theta_1 \in \Theta$, and set number of iterations $T = \lfloor n/b \rfloor$
3: **for** $t = 1, 2, \dots, T$ **do**
4: $\quad$ Draw fresh noise $\mathbf{Z}_t \leftarrow \mathcal{N}\left(\mathbf{0}, \zeta^2 \mathbf{I}_d\right)$
5: $\quad$ Initialize gradient estimate $\bar{g}_t \leftarrow \mathbf{Z}_t$
6: $\quad$ **for** $i = 1, 2, \dots, b$ **do**
7: $\quad\quad$ Update gradient estimate $\bar{g}_t \leftarrow \bar{g}_t + \frac{1}{b} \nabla_\theta \ell(\theta_t^{\mathsf{md}}, x_{t,i})$
8: $\quad$ **end for**
9: $\quad$ $\bar{g}_t \leftarrow \bar{g}_t + \mathcal{N}\left(\mathbf{0}, \zeta^2 \mathbf{I}_d\right)$
10: $\quad$ Update parameter estimate $\theta_{t+1} \leftarrow \arg\min_{\theta \in \Theta} \left\{\langle \bar{g}_t, \theta - \theta_t \rangle + \frac{L_t}{2} \|\theta - \theta_t\|_2^2\right\}$
11: $\quad$ Update aggregated parameter estimate $\theta_{t+1}^{\mathsf{ag}} \leftarrow \alpha_t \theta_{t+1} + (1 - \alpha_t)\theta_t^{\mathsf{ag}}$
12: $\quad$ Update middle parameter estimate $\theta_{t+1}^{\mathsf{md}} \leftarrow \alpha_t \theta_{t+1} + (1 - \alpha_t)\theta_t^{\mathsf{ag}}$.
13: **end for**
14: Output $\theta_{T+1}^{\mathsf{ag}}$

---

*Proof.* Let $t^*$ (resp. $i^*$) be the batch number (resp. position within the batch) where the adversary intrudes. We will show that

$$D_\infty^\delta(\theta_{t^*}, \theta_{t^*}^{\mathsf{ag}}, \theta_{t^*}^{\mathsf{md}}, \bar{g}_{t^*}, \theta_{T+1}^{\mathsf{ag}} \| \theta_{t^*}', \theta_{t^*}^{\mathsf{ag}\,\prime}, \theta_{t^*}^{\mathsf{md}\,\prime}, \bar{g}_{t^*}', \theta_{T+1}^{\mathsf{ag}\,\prime}) \leq \varepsilon$$

where the $'$ indicates variables generated from an execution on neighboring stream $X'$.

For neighboring streams $X, X'$, let $t$ denote the batch number and let $i$ denote the number of the sample within the batch where the two streams differ. We will perform case analysis around $t$.

Case 1: $t^* < t$. Here, the tuple $(\theta_{t^*}, \theta_{t^*}^{\mathsf{ag}}, \theta_{t^*}^{\mathsf{md}}, \bar{g}_{t^*})$ is identically distributed with $(\theta_{t^*}', \theta_{t^*}^{\mathsf{ag}\,\prime}, \theta_{t^*}^{\mathsf{md}\,\prime}, \bar{g}_{t^*}')$ which means we only need to prove $D_\infty^\delta(\theta_{T+1}^{\mathsf{ag}} \| \theta_{T+1}^{\mathsf{ag}\,\prime}) \leq \varepsilon$.

Observe that $\theta_{T+1}^{\mathsf{ag}}$ (resp. $\theta_{T+1}^{\mathsf{ag}\,\prime}$) is a post-processing of $(\theta_t, \theta_t^{\mathsf{ag}}, \theta_t^{\mathsf{md}}, \bar{g}_t)$ (resp. $(\theta_t', \theta_t^{\mathsf{ag}\,\prime}, \theta_t^{\mathsf{md}\,\prime}, \bar{g}_t')$), which means that we only need to bound $D_\infty^\delta(\theta_t, \theta_t^{\mathsf{ag}}, \theta_t^{\mathsf{md}}, \bar{g}_t \| \theta_t', \theta_t^{\mathsf{ag}\,\prime}, \theta_t^{\mathsf{md}\,\prime}, \bar{g}_t')$. Furthermore, the parameter estimates are generated from a post-processing of the gradient updates, it suffices to bound $D_\infty^\delta(\bar{g}_t \| \bar{g}_t')$ after line 9. Recall the Gaussian mechanism:

**Lemma C.5.** *For $\varepsilon < 1$ and function $f \colon A^n \to B^d$ on size-$n$ databases with $\ell_2$-sensitivity*

$$\Delta_2(f) = \max_{\substack{neighboring\ a,a' \in A^n}} \|f(a) - f(a')\|_2$$

*outputting*

$$f(a) + \mathcal{N}\left(\mathbf{0}, \frac{2\Delta_2(f)^2 \log(2/\delta)}{\varepsilon^2} \mathbf{I}_d\right)$$

*where $\mathbf{I}_d$ is the $d$-dimensional identity matrix, is $(\varepsilon, \delta)$-centrally private.*

For a proof, refer to Theorem A.1 in the survey of Dwork and Roth (Dwork and Roth, 2014).

Since $\ell$ is $L$-Lipschitz in $\theta$, for any $x$ we have $\|\nabla_\theta \ell(\cdot, x)\|_2 \leq L$. Thus

$$\max_{x, x' \in \mathcal{X}} \|\nabla_\theta \ell(\cdot, x) - \nabla_\theta \ell(\cdot, x')\|_2 \leq 2L.$$

Thus if we define $f(x_{t,1}, \dots, x_{t,b}) = \frac{1}{b} \sum_{i=1}^b \nabla_\theta \ell(\theta_t, x_{t,i})$, we get $\Delta_2(f) \leq \frac{2L}{b}$. By Lemma C.5, it follows that our addition of $\mathbf{Z}_t \sim \mathcal{N}\left(0, \frac{8L^2 \log(2/\delta)}{b^2 \varepsilon^2}\right)$ noise guarantees $(\varepsilon, \delta)$-privacy

Case 2: $t^* = t$. We break into more cases around $i^*$.

When $i \leq i^* < b$, the adversary only observes a change to the gradient (line 7). Specifically, the tuple $(\theta_{t^*}, \theta_{t^*}^{\mathsf{ag}}, \theta_{t^*}^{\mathsf{md}}, \theta_{T+1}^{\mathsf{ag}})$ is identically distributed with $(\theta_{t^*}', \theta_{t^*}^{\mathsf{ag}\,\prime}, \theta_{t^*}^{\mathsf{md}\,\prime}, \theta_{T+1}^{\mathsf{ag}\,\prime})$ which means it suffices to prove $D_\infty^\delta(\bar{g}_t \| \bar{g}_t') \leq \varepsilon$ after line 7. This follows from the privacy of the Gaussian mechanism.

When $i \leq b = i^*$, observe that the parameter updates are post-processing of the gradient update. We again use the privacy of the Gaussian mechanism.

When $i^* < i$, we re-use the arguments from Case 1. The noise from line 9 provide privacy.

Case 3: $t^* > t$. Observe that $\theta^{\text{ag}}_{T+1}$ (resp. $\theta^{\text{ag}}_{T+1}{}'$) is a post-processing of $(\theta_{t^*}, \theta^{\text{ag}}_{t^*}, \theta^{\text{md}}_{t^*}, \overline{g}_{t^*})$ (resp. $(\theta'_{t^*}, \theta^{\text{ag}}_{t^*}{}', \theta^{\text{md}}_{t^*}{}', \overline{g}'_{t^*})$), which means that we only need to bound $D^{\delta}_{\infty}(\theta_{t^*}, \theta^{\text{ag}}_{t^*}, \theta^{\text{md}}_{t^*}, \overline{g}_{t^*} \| \theta'_{t^*}, \theta^{\text{ag}}_{t^*}{}', \theta^{\text{md}}_{t^*}{}', \overline{g}'_{t^*})$. But the random variables in question are obtained by post-processing $(\theta_t, \theta^{\text{ag}}_t, \theta^{\text{md}}_t, \overline{g}_t)$ and $(\theta'_t, \theta^{\text{ag}}_t{}', \theta^{\text{md}}_t{}', \overline{g}'_t)$. We use the same line of reasoning as Case 1 to bound the divergence. $\square$

The main tool for our utility guarantees is bounding the variance of the noise added for privacy, analogous to the variance guarantee of Theorem 3.2. Once we have that result, the remaining analyses are essentially unchanged from their shuffle counterparts, apart from slightly different choices for parameters $b$ and $\beta$, which are straightforward algebraic consequences of the different variance guarantee. As a result, we only bound the variance.

For each time step $t \in [T]$, the gradient $\overline{g}_t = \frac{1}{b} \sum_{i=1}^{b} \nabla_{\theta} \ell(\theta_{t-1}, x_{t,i}) + \mathbf{Z}_{t,a} + \mathbf{Z}_{t,b}$, where $\mathbf{Z}_{t,a}, \mathbf{Z}_{t,b} \sim \mathcal{N}\left(\mathbf{0}, \zeta^2 \mathbf{I}_d\right)$. It is clearly an unbiased estimator and the variance satisfies

$$\mathbb{E}_{x_{t,1},\ldots,x_{t,b},\mathbf{Z}_{t,a},\mathbf{Z}_{t,b}} \left[ \left\| \frac{1}{b} \sum_{i=1}^{b} \nabla_{\theta} \ell(\theta_{t-1}, x_{t,i}) + \mathbf{Z}_{t,a} + \mathbf{Z}_{t,b} - \mathbb{E}_{x \sim \mathcal{D}} \left[ \nabla \ell(\theta_{t-1}, x) \right] \right\|_2^2 \right]$$

$$= \mathbb{E}_{x_{t,1},\ldots,x_{t,b}} \left[ \left\| \frac{1}{b} \sum_{i=1}^{b} \nabla \ell(\theta_{t-1}, x_{t,i}) - \mathbb{E}_{x \sim \mathcal{D}} \left[ \nabla \ell(\theta_{t-1}, x) \right] \right\|_2^2 \right] + \mathbb{E} \left[ \|\mathbf{Z}_{t,a}\|_2^2 \right] + \mathbb{E} \left[ \|\mathbf{Z}_{t,b}\|_2^2 \right]$$

$$= \frac{1}{b} \mathbb{E}_{x_t \sim \mathcal{D}} \left[ \| \nabla \ell(\theta_{t-1}, x_t) - \mathbb{E}_x \left[ \nabla \ell(\theta_{t-1}, x) \right] \|_2^2 \right] + \mathbb{E} \left[ \|\mathbf{Z}_{t,a}\|_2^2 \right] + \mathbb{E} \left[ \|\mathbf{Z}_{t,b}\|_2^2 \right]$$

$$= O\left( \frac{L^2}{b} + \frac{dL^2 \log(1/\delta)}{b^2 \varepsilon^2} \right).$$

The first step follows from the fact that the noise $\mathbf{Z}_{t,a}, \mathbf{Z}_{t,b}$ is independent of the data $x_{t,1}, \cdots, x_{t,b}$, and the second step follows from the independence of $x_{t,1}, \cdots, x_{t,b}$. The last step follows from $\mathbf{Z}_{t,a}, \mathbf{Z}_{t,b} \sim \mathcal{N}\left(\mathbf{0}, \zeta^2 \mathbf{I}_d\right)$ and $\zeta^2 \leftarrow \frac{8L^2 \log(2/\delta)}{b^2 \varepsilon^2}$. $\square$

