# OpenReview forum: "Shuffle Private Stochastic Convex Optimization"
_ICLR.cc/2022/Conference — ICLR 2022 Poster_

### Official Review · Reviewer_d49M · 2021-10-30

**Correctness:** 3
**Technical Novelty And Significance:** 3
**Empirical Novelty And Significance:** Not applicable
**Recommendation:** 6
**Confidence:** 4

**Main Review:**

Strengths:

-The privacy mechanism for vector summation is interesting. It differs from the standard use of additive noise (usually Gaussian or Laplace) in DP optimization. It is also somewhat novel, as most prior works were for vectors with bounded $L_1$ sensitivity instead of $L_2$ (or used privacy amplification, c.f. Girgis et al (2021)).

-The problem is interesting, considering the practical relevance of shuffle DP (as an intermediate trust model between local and central DP) and the importance of SCO. In light of the work of Erlingsson et al. (2020) and Girgis et al. (2021) on shuffle DP ERM, SCO is a natural next step.

-The bounds seem good, especially Theorem 4.9 since it shows optimality.

-The paper is well-written and the results seem correct.

Weaknesses:
-For me, the weakest link of the paper is Section 4.1 (sequentially interactive SCO). The results are suboptimal as they stand, so one wonders why they are useful. The authors do mention that in practice, sometimes only one query per user is possible, which provides some motivation for sequentially interactive algorithms. Can the authors provide references and practical examples to support this claim? I would also like to see other arguments motivating the importance of sequentially interactive protocols. In addition, the paper would be greatly strengthened by either improving the bounds in Section 4.1 (to match the existing central DP lower bounds) or adding nearly tight lower bounds for sequentially interactive algorithms to show a separation between sequential and full interactivity for this problem and establishing near optimality of their bounds. Joseph et al. (2019) seems like a relevant work for this.

-Missed related work: 1. There is a very relevant contemporaneous work that should be cited/discussed in the final version of the paper by Lowy & Razaviyayn (2021)  https://arxiv.org/abs/2106.09779. They cover a more general problem of federated learning, but setting $n=1$ and $M=N$ in their Theorem 4.1 recovers your Theorem 4.9. They use different protocols and only have fully interactive algorithms. They also use acceleration and smoothing for some of their results. I don't hold this one against the authors since it is concurrent, but a discussion of this work should be added to the Related Work section of the final version of the paper. 2. Erlingsson et al. (2020) ("ESA Revisited..") should also be referenced with Girgis et al (2021) in the context of shuffle DP ERM. 3. Smith et al. (2017) ("Is interaction necessary...") uses acceleration (Theorem 20), as does Lowy & Razaviyayn (2021)--these should be discussed, and the sentence "we are not aware of previous private SCO results relying on acceleration" in Contribution 3 should be removed.

-Intro: $O(\sqrt{d}/\epsilon \sqrt{n})$ is optimal in the local model is true, but I think Duchi et al (2013) only proves it for sequentially interactive pure DP algorithms, right? However, Lowy & Razaviyayn (2021) show that the same optimal loss bound holds for approximate DP fully interactive algorithms.

-Definition 2.7 seems unnecessarily abstract; would be good to explain intuitively

-"allow the analyzer to adaptively choose which subset of users to query.." is not the main/important difference between full and sequential interactivity; sequential algorithms also choose which subset of users to query in each round (but not adaptively). I would either delete this part of the sentence or emphasize "adaptively"; the main distinction is in the second part of the sentence (repeated querying).

-Lemma 3.1: y is never explicitly defined; you should define it and say it lives in $[0,1]^{g+b}$; "$P_{1D}(S(y))$ is an unbiased estimate..." is not technically correct (as it is a tuple)--I think you mean "$A_{1D}(S(y))$ is an unbiased estimate..."?

-Algorithms 3 and 5: it looks like you are doing gradient ascent instead of descent

-Section 4.1 initial paragraph: include $\epsilon$ in these loss bounds

-Restriction on epsilon should be included in statements of all theorems in Section 4. You require $\epsilon \leq 1$ for privacy of your vector summation protocol, right? Also, for advanced composition in Theorem 4.9, you need some restriction (even if very mild and possibly subsumed by the restriction needed for vector summation) to bound $e^{\epsilon'} \leq 2 \epsilon'$. Also, I think you may need to shrink the inputs to your vector summation protocol by constant factor (something like $2\sqrt{2}$) to get exactly $(\epsilon, \delta)$-DP in Theorem 4.9.

-Clarify wording in "combining it with acceleration to attain a better guarantee": the guarantee is not better than that attained in Bassily et al. (2019).

-"optimization trajectory": what does this mean?

-Theorem 4.9: I think you are missing a $lambda$ in the denominator of the second term? (Or the units are wrong.); also, you should include runtime and/or communication complexity guarantees (and do the same for results in Section 4.1.)

-Proof of Lemma 3.1. Clarify/highlight where/how shuffling is used to provide privacy (if at all). It wasn't clear to me. Would same privacy guarantee hold without shuffling? Also, explain why $Var(\eta_1) \leq 1/4.$

**Summary Of The Paper:**

The paper provides shuffle DP algorithms and population loss upper bounds for stochastic convex optimization (SCO). They include both sequentially interactive and fully interactive algorithms. Their fully interactive upper bounds match the central DP lower bounds. They propose a private protocol for vector summation that serves as the privacy workhorse of their algorithms; they combine this protocol with variants of (accelerated) gradient descent and SGD to obtain their results.

**Summary Of The Review:**

The paper is interesting, well-written, and seems correct. It is a solid work, but would be greatly strengthened by providing *tight* bounds for sequentially interactive algorithms and/or strengthening the motivation for sequentially interactive algorithms. Additionally, there are a number of minor issues (described above) that should be addressed. I consider the paper marginally below acceptance threshold in its current form, but would be open to recommending acceptance upon satisfactory revision.

__

Edit: The authors have addressed my concerns and revised the manuscript. I recommend acceptance.

---

> ### Author Response · Authors · 2021-11-12
> **Response to reviewer d49M**
>
> Thanks for the detailed evaluation! We recap the primary concerns and provide responses below. For comments not addressed below, we agree with the suggestion and will implement the change in the next draft. If desired, we would be happy to upload a rebuttal revision with these completed changes later in the discussion period.
>
> > Can the authors provide references and practical examples to support [the analysis of sequential interactivity]?
>
> The main advantage of sequentially interactive algorithms over fully interactive ones is that sequentially interactive algorithms do not require participants to remain active over the full duration of a protocol. Requiring users to participate in many repeated steps can be difficult in practice. A few references for this:
>
> "... the DP guarantees require very specific sampling or shuffling schemes assuming, for example, that each client participates in each iteration with a fixed probability. While possible in theory, such schemes are incompatible with the practical constraints and design goals of cross-device FL protocols" (BKMTT NeurIPS 20, arxiv.org/abs/2103.00039)
>
> "... user devices can go offline or otherwise become unreachable — and so it may not be possible to return to a previously queried user and pose a new query … [this] difficulty motives the study of sequentially interactive protocols" (JMNR FOCS 19, arxiv.org/abs/1904.03564)
>
> "When the selected subset of users is uniformly sampled from the underlying population of users, the privacy gains of this sampling procedure can be analyzed using the analytical moments accountant method … [h]owever, such a sampling procedure is nearly impossible in practice. This is because (a) the population of devices may not be known to the service provider, and (b) the specific subset of available devices can vary dramatically over time." (KM+ 2019, arxiv.org/abs/1912.04977v1)
>
> The sequential shuffle model is also closely related to pan-privacy, for which we derive similar results (see Appendix C).
>
> > The paper would be greatly strengthened by either improving the bounds in Section 4.1 (to match the existing central DP lower bounds) or adding nearly tight lower bounds for sequentially interactive algorithms
>
> We agree that lower bounds would help complete the understanding of shuffle private SCO. Please see the response to reviewer tikt for a short discussion of this question (this response has hit the character limit).
>
> > [Please discuss] Lowy & Razaviyayn (2021) …
>
> Thanks for the reference! We were not previously aware of this paper, which appeared on Arxiv shortly after our manuscript. We would be happy to cite and discuss it. As you mention, the main overlap between the two works is the fully interactive results. A relevant distinction is that LR21's protocol assumes users can send real vectors  to the server. Adapting algorithms that require this assumption, particularly in distributed settings, is nontrivial (CKS, NeurIPS 20 arxiv.org/abs/2004.00010; KLS, NeurIPS 21 arxiv.org/abs/2102.06387). In contrast, our protocols are entirely discrete. We will also correct our discussion of previous shuffle private ERM results and private SCO results using acceleration.
>
> > "optimization trajectory".
>
> This is shorthand for how loss and the distance to the optimum evolve.
>
> > [Y]ou should include runtime and/or communication complexity guarantees
>
> Since the focus of SCO is typically the loss guarantees, we omitted runtime and communication bounds (see discussion at the end of Section 3.1). However, they are easy to derive: each user in the vector summation protocol sends $d(g+b) \approx d(\Delta\sqrt{n} + \log(1/\delta)/\varepsilon^2)$ bits, which the analyst post-processes in time $O(dn)$. The corresponding bounds for the optimization algorithms then follow from the number $T$ of calls to the vector summation protocol and the number of times each user participates. We think the current presentation is simplest for highlighting the loss guarantees, but if desired, we can add runtime and communication as well.
>
> > Proof of Lemma 3.1. Clarify/highlight where/how shuffling is used to provide privacy (if at all). It wasn't clear to me. Would same privacy guarantee hold without shuffling? Also, explain why $Var(\eta_1) \leq ¼$.
>
> It is true that without the shuffler, there is a small degree of local privacy: each user generates b bits of Bernoulli noise alongside g encoded bits. An adversary’s view reduces to the binomial mechanism but note that the variance without shuffling (bp(1-p)) is much smaller than the variance with shuffling (nbp(1-p)). This means privacy of the shuffle protocol is stronger than the privacy of the randomizer on its own. $Var(\eta)_1) \leq ¼$ because $\eta_1$ is a Bernoulli random variable, which always has variance at most $¼$.

---

> > ### Comment · Reviewer_d49M · 2021-11-15
> > **Reply to authors**
> >
> > Thank you for the detailed response. I believe you have addressed my main concerns. I still think it would be nice to include these communication complexity/runtime bounds somewhere in the paper or appendix. It would be great if you could share the final revised version once it is ready before discussion period ends/final ratings are due.

---

> > > ### Author Response · Authors · 2021-11-17
> > > **Revised version**
> > >
> > > We have uploaded a second submission draft, featuring the following changes:
> > >
> > > 1. Expanded justification of sequential interactivity at the beginning of Section 1.1 and after Definition 2.8.
> > >
> > > 2. Removed claim that acceleration has not previously been used in private SCO (item 3 in Section 1.1).
> > > 3. Added discussion of "Encode, Shuffle, Analyze Revisited … " and "Private Federated Learning Without a Trusted Server … " in Section 1.2.
> > >
> > > 4. Clarified text after Definition 2.7.
> > >
> > > 5. Added explicit definition of $\vec{y}$ to beginning of Section 3.1.
> > >
> > > 6. Added $\varepsilon$ to bounds at the beginning of Section 4.1 and $\varepsilon$ assumption in all SCO statements.
> > >
> > > 7. Clarified "better guarantee" in discussion before Theorem 4.3, added $\lambda$ in Theorem 4.9.
> > >
> > > 8. Added discussion of communication and runtime at the end of Section 3.1. Note that our original statement about the scalar sum communication complexity dependence on $\Delta$ was incorrect: for the previous version, the correct dependence for each user is $\Delta\sqrt{n} + \Delta^2\log(1/\delta)/\varepsilon^2$ bits. At the same time, this communication dependence on $\Delta$ can easily be removed entirely by having users scale their inputs down by $\Delta$, using the current protocol as if $\Delta = 1$, then scaling the output up by $\Delta$ at the end. The result is still unbiased and has the same variance guarantee. We have added a short
> > > discussion explaining this.
> > >
> > > 9. Added clarification about Bernoulli random variable variance in proof of Lemma 3.1 in Appendix.

---

> > > > ### Comment · Reviewer_d49M · 2021-11-18
> > > > **two more (small) things**
> > > >
> > > > Thank you for the changes--this version looks much better to me. One thing is that the citation of Duchi et al (2013) for optimal rate in local model (last sentence of paragraph 2) still seems incorrect or incomplete, since they only address pure DP and sequential algorithms. Can you add Lowy & Razaviyayn (2021) there? They address approximate DP and fully interactive, showing that the stated rate (or rather, gap from the non-private rate) is indeed still optimal in the problem setting you consider in your paper.
> > > >
> > > > Also, could you also please explicitly write out the full runtime for the algorithm for clarity (i.e. runtime = T*(per iteration runtime you derived) = ...)? If space constraints are an issue, I would have no qualms about moving the runtime paragraph to an Appendix and including a pointer (e.g. you could state the total runtime in the main body to save space and move the derivation paragraph to appendix).

---

> > > > > ### Author Response · Authors · 2021-11-19
> > > > > **re: two more (small) things**
> > > > >
> > > > > > [T]he citation of Duchi et al (2013) for optimal rate in local model (last sentence of paragraph 2) still seems incorrect or incomplete, since they only address pure DP and sequential algorithms. Can you add Lowy & Razaviyayn (2021) there? They address approximate DP and fully interactive, showing that the stated rate (or rather, gap from the non-private rate) is indeed still optimal in the problem setting you consider in your paper.
> > > > >
> > > > > The new updated submission modifies the sentence as follows: "In the local model, the optimal private loss term is $O(\sqrt{d}/(\varepsilon\sqrt{n}))$ for sequentially interactive protocols (DJW13) and a class of compositional fully interactive protocols (LR21)."
> > > > >
> > > > > > Also, could you also please explicitly write out the full runtime for the algorithm for clarity (i.e. runtime = T*(per iteration runtime you derived) = ...)? If space constraints are an issue, I would have no qualms about moving the runtime paragraph to an Appendix and including a pointer (e.g. you could state the total runtime in the main body to save space and move the derivation paragraph to appendix).
> > > > >
> > > > > The new updated submission expands the communication and runtime guarantees in a new Section B.3 in the Appendix, now referenced at the beginning of Section 4.

---

> > > > > > ### Comment · Reviewer_d49M · 2021-11-19
> > > > > > **Updated my score**
> > > > > >
> > > > > > I feel good about the revised version you have provided and have changed my score to recommend acceptance.

---

### Official Review · Reviewer_6c6H · 2021-11-02

**Correctness:** 4
**Technical Novelty And Significance:** 3
**Empirical Novelty And Significance:** Not applicable
**Recommendation:** 8
**Confidence:** 3

**Main Review:**

The main strength of this paper is that, to the best of my knowledge, it is the first paper to show that there is a non-central DP setting (the full shuffle-DP setting described earlier) in which one can achieve the same excess population bounds as are possible in the central DP setting for SCO. The authors also show that in the weaker sequential model, one can still get improved excess population loss compared to the local setting. In summary, these results nicely and newly fill the gap between what is possible in the (non-shuffled) local and central settings for private SCO. I also think it is very nice that the same vector sum estimator is able to be slotted into a variety of different algorithms, and view this as a strength of the paper - it is possible that for other SCO algorithms which have been tried in the central DP literature that improve over SGD under various assumptions, one can slot in the vector sum estimator of this paper and get shuffle-DP error bounds that are not much worse "for free". If so I could see this paper having a sizeable impact on both the theoretical literature, and on how models are trained using DP in practice. As far as the writing itself, I felt the paper was nicely organized and easy to read. The authors build up the results nicely, progressing slowly from scalar sums, to vector sums, to algorithms using vector sums to arrive at their main results, and the analysis is intuitively presented. The authors also do a good job in the introduction with explaining the setting and comparing to past results to highlight the novelties of their paper in what is a quickly growing area.

I don't feel the paper has any real major weaknesses, but for completeness sake I'll address some parts of the paper that may be viewed as weaknesses. For one, there is no experimental work in the paper to show how well the algorithms do in practice. However, in some sense experiments may not be necessary - the only real difference from DP-SGD from a utility perspective is the noise being added to the gradients, which is from a near-normal distribution, so if my understanding is correct it is likely the performance of the algorithms in this paper would compare to the performance of DP-SGD but with a different scale of noise depending on the dimension/number of samples. Another possible weakness is that once one has the vector sum estimator, which follows from a composition theorem constructed by the authors, many of the SCO results are just "plug-and-play" or variants of existing analyses, i.e. most of the technical novelty is in the vector sum estimator. However, I wouldn't mark down the paper for this aspect; I think the vector sum estimator is by itself a very nice contribution for the reasons mentioned before, and I think it is also a nice "story" that in the shuffle model, we can apply algorithms/analyses that are very similar to what is used in the central model, just with added variance due to the local randomizers.

**Summary Of The Paper:**

This papers considers stochastic convex optimization (SCO) in the shuffle-DP model. In particular, the authors consider two models for shuffle-DP: A "sequential" model where the analyzer operates in rounds, and in each round a new set of users participate in a local-DP protocol, and a new "full" model in which the analyzer can request a specific subset of users to participate in each round, which allows users' data to be queried more than once. The authors show that in the full model, one can retrieve excess population loss bounds matching the best possible bounds in the central DP setting. Furthermore, they show even the sequential model offers improved excess population bounds over the best possible bound of sqrt(d/n) in the local setting.

The key technical ingredient is a shuffle-DP randomizer for estimating the sum of vectors, each stored by a user, when the vectors have bounded $\ell_2$-sensitivity (which corresponds to a Lipschitz assumption on each loss function in SCO), in contrast with existing work which focused on $\ell_1$-sensitivity. The authors build off a scalar sum estimator and then use a composition theorem to prove privacy of the vector sum estimator as composition of scalar sum estimators, and then bound the variance of this estimator, and then use it to compute batch gradients in existing optimization algorithms such as gradient descent, stochastic gradient descent, and the accelerated stochastic approximation algorithm. For some of these results, a variance bound on the batch gradient immediately gives an excess empirical loss bound. In some other settings there is no plug-and-play error bound, so instead they need appeal to arguments/reductions similar to those appearing before in the central DP SCO literature, but with some elements specialized to their setting.

**Summary Of The Review:**

Overall I greatly enjoyed reading the paper and think the results have a lot of potential for impact, both theoretically and practically. In particular the vector sum estimator provided by the authors is a tool that I could easily see being used by other papers to establish results in the shuffle-DP setting, and even just focusing on the error bounds, the optimal excess loss in the full shuffle model is quite a nice result, and the results for the sequential shuffle model also nicely improve on the results for SCO in the standard local DP model. In addition, the paper is nicely written and easy-to-read, and in turn ready for presentation at a major conference. For these reasons I recommend accepting the paper.

---

> ### Author Response · Authors · 2021-11-12
> **Response to reviewer 6c6H**
>
> Thanks for the positive evaluation! We would be happy to answer any further questions.

---

### Official Review · Reviewer_tikt · 2021-11-02

**Correctness:** 4
**Technical Novelty And Significance:** 4
**Empirical Novelty And Significance:** Not applicable
**Recommendation:** 8
**Confidence:** 3

**Main Review:**

Strength:
- The proposed interactive models are reasonable extensions of the non-interactive shuffle privacy model.
- The paper reveals non-trivial bounds on the excess errors of the shuffle private stochastic convex optimizations.
- The error bound in the convex and fully interactive case matches the known best error in the central model.

Weakness:
- The tightness of the bounds in the sequential model is unclear.


Overall, I recommend acceptance. This paper is well-written and has significant advances in the field of shuffle differential privacy.

The sequential and fully interactive models in the shuffle privacy are natural and reasonable extensions from the non-interactive model as well as the interactive models in the local privacy. I guess these models are acceptable even in real-world situations (particularly the sequential interactive model).

The authors introduce a novel analysis for the scalar sum, revealing the instance-specific privacy guarantee. This technique enables us to obtain the lower dimensional dependency on the error of the vector sum protocol.

The excess error bounds on the shuffle private stochastic convex optimization seem to be significant contributions to the related fields.

One possible drawback is the unclarity in the tightness of the bounds. In all cases, the first terms in the error bounds match the lower bound on the non-private first-order stochastic convex optimization. The second terms in the strongly convex and smooth case in the sequential model and all cases in the full model match the known best result in the central model. It is likely to be tight in these cases. The other cases, however, are unclear. The discussions about the tightness of these bounds are helpful for the reader.

**Summary Of The Paper:**

The authors deal with the stochastic convex optimization problem under the shuffle differential privacy constraint. They extend the existing non-interactive shuffle privacy model to the interactive ones, sequential and full interactive models. Under these interactive models, they propose the shuffle private SCO algorithms with the analyses of the excess error bounds. To construct their algorithms, they introduce a novel shuffle private estimation mechanism for the sum of the vectors.

**Summary Of The Review:**

Overall, I recommend acceptance. This paper is well-written and has significant advances in the field of shuffle differential privacy. One possible weakness is unclearly in the tightness of bounds in the sequential model.

---

> ### Author Response · Authors · 2021-11-12
> **Response to reviewer tikt**
>
> Thanks for the positive evaluation! We agree that lower bounds would help complete the understanding of shuffle private SCO. One obstacle to proving such lower bounds is that the tight lower bound instances for central and local differential privacy rely on simple linear loss functions, which then reduce to known lower bounds for estimating a single vector (see Section 5 in BST, FOCS 14 https://arxiv.org/abs/1405.7085). For shuffle privacy, it is easy to construct a noninteractive protocol that achieves the same loss guarantee as the central model for these loss functions, for example by applying our vector summation protocol. Our loss guarantees for arbitrary convex losses, which interpolate between the tight central and local guarantees, suggest that a more complex (and perhaps necessarily adaptive) lower bound instance is necessary to achieve the stronger lower bound we conjecture exists. Constructing such an instance may be an interesting direction for future work.

---

### Official Review · Reviewer_Frn5 · 2021-11-03

**Correctness:** 4
**Technical Novelty And Significance:** 2
**Empirical Novelty And Significance:** Not applicable
**Recommendation:** 6
**Confidence:** 4

**Main Review:**

The main contribution of the paper is the non-interactive mechanism to compute \ell_2 norm of a given vector. This in turn builds on the mechanism given by Balcer and Cheu for privately scalar sum. The algorithm for SGD is pretty straightforward and the analysis is also standard in the literature. In total, the "proposed" new contribution of the paper is the scalar sum protocol.

The scalar sum protocol is a generalization of Balcer and Cheu. The difference between the two protocols is that the randomizer of this protocol also uses how many random copies of both 0s and 1s should be sent. Each of these numbers is itself a random variable. The sampling probability for both Bernoulli and Binomial samples itself depends on the input! I found this rather surprising that they were able to prove privacy for such a protocol. However, I feel that the protocol itself reduces to the binomial mechanism proposed in Dwork et al. 2006 when we set $g \approx \sqrt{n}$. I would like to know what is the difference? In particular, when all $x$ are close to 0, then step 5 with high probability would not add anything to $\bar x$ and all we are left with is the Binomial mechanism. In particular, step 5 is more of a randomized rounding and I can see as post-processing. I do not see why it helps in privacy, and if it does not, then the mechanism is just the Binomial mechanism of "Our data, ourselves: Privacy via distributed noise generation.” If there is no difference then there is nothing new in this paper!

Apart from this, I have some serious questions about the applicability of this paper:
1. The choice of $g$ defines how many bits everyone has to send. That means, that every user has to send $\sqrt{n}$ bits, where $n$ are the participating parties. Now, in the setting where the shuffle model would be used in practice is where we try to train an ML model using private federated learning. There, in practice, we will never know in advance what is the value of $n$. How would the user know how many bits he has to send?
2. Second, and this is also a glaring problem with Balcer and Cheu, is that the randomizer sends a certain number of copies of 1s and 0s. This number is a random variable that depends on the privacy noise. An architecture for shuffler would need to know what is the input size to design the permutation. Oblivious permutation itself takes $O(m \log m)$ time, where $m$ is the input, but it crucially depends that we know the number $m$. If you want to implement a permutation where the size of the permutation matrix is a random variable, you would need to impose some more trust assumptions. This has been studied in the security and crypto community and easy deanonymization is possible. That is one reason why the IKOS 2006 assumed $m$ is known.
3. The authors give an algorithm and analyze it under various settings for SGD. However, they fail to compare it "Practical and Private (Deep) Learning Without Sampling or Shuffling" (ICML 21) That paper does a very good job of arguing why sampling and shuffling are problematic in large-scale deployment. However, more importantly, they give an algorithm that meets the bounds one get by shuffling or sampling (up to $\log n$ factor; which I think can be improved using better data structure). Their privacy guarantee more crucially does not depend on convexity, a significant departure from previous work in this area starting with Song et al. What this paper throws under the rug is the fact that epsilon when you are training or using convex optimization would never be less than $1$. So, after getting the terms which is true for any epsilon, they always make the assumption that $\epsilon$'s are small! In practice, using Honaker's improvement on binary tree mechanism, I would not be surprised that the ICML 21 paper would be better than the submission in all axis once the privacy budget is reasonable for where we use any private convex optimization.
4. Finally, the proof of the composition theorem is essentially the proof of composition theorem from "Boosting and DP" paper. The current paper can appeal to Chernoff while the Boosting and DP paper required Azuma as they had to deal with a martingale (and their composition was more general).

I have a few more small questions that I will leave to ask during the reviewer-author engagement.


Post author's response phase: The authors have answered most of my questions in a timely manner. I really appreciate their response. It helped me better understand the paper. I would ideally like to give a paper an accept (a rating of 7), which I cannot find. So, I would request the AC to consider this remark over the score given in the Recommendation section.

**Summary Of The Paper:**

The paper studies stochastic convex optimization in the shuffle model of privacy. They claim a new non-interactive algorithm for privately computing \ell_2 norm, which would be useful in the stochastic gradient descent algorithm.

**Summary Of The Review:**

The paper studies stocastic convex optimization in shuffle model; however, to my understanding, most of the results are known or are folklore. Also, they fail to compare with the most relevant and recent already published work.

I have studied the proof and they are correct (more so, because most of them follow the standard ideas or the results follow from previous works).

---

> ### Author Response · Authors · 2021-11-12
> **Response to reviewer Frn5**
>
> Thanks for the detailed evaluation of our paper! We recap the concerns and provide responses below:
>
> > I feel that the [scalar sum] protocol itself reduces to the binomial mechanism proposed in Dwork et al. 2006 … If there is no difference then there is nothing new in this paper … most of the results are known or are folklore
>
> We agree that the scalar sum protocol relies on the binomial mechanism as its primary primitive. However, we note that 1) its analysis requires adaptation of existing results for the binomial mechanism to obtain the unbiased shuffle-private variant here, and 2) the scalar sum protocol is only the first building block for our eventual (unbiased) vector sum protocol and the resulting stochastic convex optimization (SCO) guarantees, both of which require further new analyses and are themselves new results in the literature.
>
> > The choice of $g$ defines how many bits everyone has to send … how would the user know how many bits he has to send?
>
> The protocol requires the analyst to set, roughly, $g > \sqrt{n}$. This means that a conservative upper bound on the number of users $n$ suffices. We believe that a conservative upper bound on the number of users is a reasonable assumption, particularly since the communication cost only scales with its square root.
>
> > the randomizer sends a certain number of copies of 1s and 0s … [a]n architecture for shuffler would need to know what is the input size to design the permutation
>
> The proportions of 0s and 1s in the message are randomized, but the message length is not randomized: it is always $g + b$ bits, where $g$ and $b$ are set by the analyst before the protocol begins.
>
> > "Practical and Private (Deep) Learning Without Sampling or Shuffling" (ICML 21) … give[s] an algorithm that meets the bounds one get by shuffling or sampling
>
> The referenced paper provides an algorithm for SGD that satisfies central differential privacy and obtains a $d^{1/4}/\sqrt{n}$ SCO (population) loss guarantee in a single pass over the data. In contrast, our paper provides an algorithm that satisfies the separate, stronger notion of shuffle differential privacy and obtains a $d^{1/3} / n^{2/3}$ SCO loss guarantee in a single pass over the data. We agree that the referenced paper's adaptability to nonconvex losses and ability to avoid sampling or shuffling can be useful in practice. However, the focus of our paper is shuffle private SCO, and we provide stronger loss guarantees in a stronger privacy model. The results in our paper do not follow from the results in the referenced paper.
>
> > What this paper throws under the rug is the fact that epsilon when you are training or using convex optimization would never be less than 1.
>
> $\varepsilon \leq 1$ is a common assumption in the literature on differentially private SCO (e.g. BST, FOCS 14 https://arxiv.org/abs/1405.7085, BFTT, NeurIPS 19 https://arxiv.org/abs/1908.09970, FKT, STOC 20 https://arxiv.org/abs/2005.04763). This is because the goal of private SCO is to obtain asymptotic loss guarantees, and the exact constant assumptions are not the focus.
>
> > [T]he proof of the composition theorem is essentially the proof of composition theorem from "Boosting and DP" paper
>
> We agree that the two proofs have high-level similarities. However, our result differs by allowing different privacy guarantees for each coordinate. This generalization is necessary to prove our final vector aggregation result, which uses the $\ell_2$ sensitivity of the entire vector.

---

> > ### Comment · Reviewer_Frn5 · 2021-11-16
> > **Response to rebuttal**
> >
> > Unbiasedness: Making binomial mechanism unbiased is not very hard if I understand it correctly. Given that it is the second mechanism that was proposed in DP literature, it has been used more or less as folklore.
> > Binomial to vector sum: For vector sum, I do not see how it is very different from doing co-ordinate wise binomial mechanism + recentering? The former you need in most applications of binomial mechanism (I cannot claim in every mechanism, but at least the one I have seen in practice).
> >
> > Conservative upper bound. It is a reasonable assumption to have a conservative upper bound on $n$ given that $n$ fluctuates in practice because of many reasons -- phone not being plugged in for charging at the time of model collection, etc; my critique was that scaling of communication with $\sqrt{n}$. If you talk to any engineer who had tried to deploy FL system and say you are trying to do learning using keyboard data from smartphones, in practice, conservative bounds on $n$ would be in the orders of millions. Given that language models are so huge, you already are pushing a 1000 times or more factor overhead on communication. That is HUGE in practice and you will start seeing dropouts and consequently no data.
> >
> > The number of bits to shuffler: Algorithm 1 clearly mentions in line 7 that the reported number of bits is a random variable. I am unsure what the authors are trying to tell me here. Since Algorithm 1 is used as subroutine in vector sum and SGD, why would that not be translated to higher-order routines?
> >
> > The ICML paper discusses in detail the issue with shuffler. A partial group of the same authors also used random check-ins in their NeurIPS paper. There is a reason they discuss shuffler and why their algorithm beats shuffling-based learning. While their algorithm is presented in the central setting, it is not hard to modify it to the distributed setting. All the steps have a very simple distributed analogue. Starting with the work of Bassily et al. (NeurIPS 2019), one setting we also care is when $n \approx d$, where the submitted version matches the results in the ICML paper. Further, if you are deploying large scale models in the Federated setting (which I believe your motivation is since you use shuffling), at many updates, the real $n$ is smaller than $d$.
> >
> > Yes, $\epsilon \leq 1$ is a standard thing to do in theoretical SCO papers mainly because they use amplification by sampling and the Li et al. correction (or recent work by Yu-Xiang, Balle, Kaisivishwanathan, etc) have shown the dependency to scale as $e^{\epsilon} -1$. On the other hand, for shuffling, Hiding among clones paper went to that extreme to get the constant right because of this very dependence on exponential form. In particular, I do not know anywhere you would use DP learning in practice, you would have $\epsilon <1$. This was not though the reason to ask the question. The reason was that I felt that the authors went to a length using the right form all the way before moving to approximation just to make the result look nicer!
> >
> > Proving privacy for each coordinates: how is it still different? Can you point to exact place where if we had not used DRV's proof coordinate (or for that matter Kairouz, Oh, Viswanath's paper that does it for different epsilons as well), we would have got things different?

---

> > > ### Author Response · Authors · 2021-11-17
> > > **re: Response to rebuttal**
> > >
> > > Thanks for the prompt reply!
> > >
> > > > It is a reasonable assumption to have a conservative upper bound on $n$ … my critique was that scaling of communication with $n$
> > >
> > > Thank you for clarifying; our interpretation of the original comment was that the concern was over knowing $n$. We agree that there are federated learning problems where the relevant $n$ is large enough for $\sqrt{n}$ communication to become challenging. As noted at the end of Section 3.1 in the paper, 1) existing tools offer lower communication at the cost of a more complex algorithm, and 2) we opted for the simpler protocol with higher communication because we focus on error guarantees. We respectfully suggest that this is a common pattern, particularly in shuffle privacy: one paper contributes the basic algorithmic idea and error guarantees, and subsequent work makes focused improvements to its practicality and communication (e.g. real summation in arxiv.org/abs/1808.01394 and then arxiv.org/abs/1906.09116; or distinct elements in arxiv.org/abs/2004.09481 and then arxiv.org/abs/2009.09604). We suggest that this paper is a useful first step in this direction. We have also uploaded a new submission draft that states communication and runtime guarantees at the end of Section 3.1.
> > >
> > > > The number of bits to shuffler: Algorithm 1 clearly mentions in line 7 that the reported number of bits is a random variable
> > >
> > > In Algorithm 1, the number of 1s sent and the number of 0s sent are both random variables. However, the reported number of bits is the sum of these numbers, which is not a random variable, as all terms in the sum cancel, except for $g + b$.
> > >
> > > > The ICML paper discusses in detail the issue with shuffler … and why their algorithm beats shuffling-based learning. While their algorithm is presented in the central setting, it is not hard to modify it to the distributed setting … Starting with the work of Bassily et al. (NeurIPS 2019), one setting we also care is when $n \approx d$, where the submitted version matches the results in the ICML paper
> > >
> > > If we understand correctly, this comment raises two points. First, there are practical considerations for implementing shuffle privacy. Second, there are the SCO bounds. We suggest that the first question is out of scope for this paper, as it can be asked of any of the (now many) papers about shuffle privacy. For the second question, taking for granted that it is easy to adapt their results for shuffle privacy: as mentioned in our original response, the ICML paper offers a $d^{1/4}/\sqrt{n}$ SCO guarantee (section 4.3 in their paper), and this is strictly worse than our $d^{1/3}/n^{2/3}$ guarantee (Theorem 4.7 in our paper) in the $n \approx d$ setting referenced here (and the canonical setting in the private SCO literature). Our fully interactive results further improve on this guarantee. Given the different guarantees, could you elaborate on how our paper only "matches the results in the ICML paper"?
> > >
> > > > … I felt that the authors went to a length using the right form all the way before moving to approximation just to make the result look nicer!
> > >
> > > If the concern is that the assumption on epsilon is not clear in the SCO results, we're happy to add it explicitly in the theorem statements, as done in the updated submission draft.
> > >
> > > > Unbiasedness: Making binomial mechanism unbiased is not very hard if I understand it correctly ... it has been used more or less as folklore ... [f]or vector sum, I do not see how it is very different from doing co-ordinate wise binomial mechanism + recentering? The former you need in most applications of binomial mechanism … [c]an you point to exact place where if we had not used DRV's proof coordinate (or for that matter Kairouz, Oh, Viswanath's paper that does it for different epsilons as well), we would have got things different?
> > >
> > > At a high level, while the essential application of advanced composition is not new to differential privacy, we disagree that the overall construction is folklore. Versions of its basic components exist in the literature, but putting them together and adapting them for shuffle privacy requires some care (in particular, we expect that swapping in KOV15's result would actually be more cumbersome than the current proof). Technical novelty is always at least somewhat subjective, but we refer to the other three expert reviews ("a novel analysis for the scalar sum", "I also think it is very nice that the same vector sum estimator is able to be slotted into a variety of different algorithms, and view this as a strength of the paper", "[t]he privacy mechanism for vector summation is interesting") and suggest that this at least contradicts the "folklore" label. That said, if provided with references to existing applications of the binomial mechanism that are indeed similar to ours, we would be happy to discuss them in the paper.

---

> > > > ### Comment · Reviewer_Frn5 · 2021-11-17
> > > > **re: discussion**
> > > >
> > > > I agree that existing protocols have lower communication cost at the expense of being more complicated. I can see where it is a problem in practice; implementing such complicated algorithm at large scale can be challenging, but if I understand the motivation behind shuffle model, it was to keep algorithms simple and achieve the central DP guarantee even though we are in the federated setting. If the communication scales like $\log(n)$, then is still fine, but once it starts to scale like $\sqrt{n}$, it becomes completely impractical. That is why I like Balle et al. has just one bit of communication. Balcer et al. was a theoretically beautiful paper, but its communication cost scaled with the universe size. That is why I like Chen et al. that got it down to $O(\log (|\mathcal{X}|))$. I mean if we have so much communication, why not just run secure MPC! If the authors are fine their paper being more theoretical importance, then it is a different point altogether. (I will have a look at the updated version asap. Thanks)
> > > >
> > > > Great! Thanks for correcting my oversight. This is good; I still feel that $\sqrt{n}$ communication per co-ordinate is going to be a huge overhead on any architecture that supports shuffling, but I can move it aside by making a case that the paper is of theoretical importance and as the authors mentioned more for initiating the study in this direction.
> > > >
> > > > If I remember correctly, for general, Lipschitz convex losses, the population risk of the ICML paper using FTRL is the same as Bassily et al. (which is optimal) up to a logarithmic factor. Thanks for pointing to your fully interactive part. It would be great to add a note about the pros and cons of interactivity. I think it will make the paper more accessible to a wider audience.
> > > >
> > > > I think it would definitely benefit the paper if you mention the dependence on $\epsilon$ and then use a corollary stating the situation when $\epsilon <1$. I am emphasizing this point more and more because in practice $\epsilon>1$ and practitioners often try to compare the utility guarantee from the main Theorem statement and when it does not meet the expectation, they cast the paper aside.
> > > >
> > > > For the composition theorem, I would really appreciate it if the authors can point me to the exact place where things differ and where adapting Dwork et al. (or many other works following that) would not work. I am just trying to understand the paper and want to end this review process at the stage where if I accept the paper, I can defend it as my own. For example, the dependence of different $\epsilon_1, \cdots, \epsilon_k$ is mentioned implicitly in John Duchi's notes (https://web.stanford.edu/class/stats311/lecture-notes.pdf) and Corollary 2.3 in this paper (https://arxiv.org/pdf/1912.04042.pdf).

---

> > > > > ### Author Response · Authors · 2021-11-19
> > > > > **re: discussion**
> > > > >
> > > > > Thanks again for the prompt response!
> > > > >
> > > > > > If I remember correctly, for general, Lipschitz convex losses, the population risk of the ICML paper using FTRL is the same as Bassily et al. (which is optimal) up to a logarithmic factor.
> > > > >
> > > > > Perhaps there's some confusion over which Bassily et al. paper the ICML paper is referring to? The ICML paper emphasizes that its SCO results match those of Bassily, Smith, Thakurta 14 (arxiv.org/abs/1405.7085). For example, in the ICML paper's Section 1.2, they state that "[f]or general Lipschitz convex losses, the population risk for DP-FTRL in Theorem C.5 is same as that in [BST14, Appendix F] (up to logarithmic factors), but the advantage of DP-FTRL is that it is a single pass algorithm". However, the $d^{1/4}/\sqrt{n}$ SCO guarantee in BST14 is suboptimal; the optimal $\sqrt{d}/n$ result only comes from Bassily, Feldman, Talwar, Thakurta 2019 (arxiv.org/abs/1908.09970). The ICML paper does not match this later guarantee. Our sequentially interactive $d^{1/3}/n^{2/3}$ SCO guarantee lies between the BST14 and BFTT19 results.
> > > > >
> > > > > > Thanks for pointing to your fully interactive part. It would be great to add a note about the pros and cons of interactivity. I think it will make the paper more accessible to a wider audience.
> > > > >
> > > > > We agree, and reviewer d49M also requested this change. Please see the discussion added at the beginning of Section 1.1 and after Definition 2.8.
> > > > >
> > > > > > I think it would definitely benefit the paper if you mention the dependence on $\varepsilon$ and then use a corollary stating the situation when $\varepsilon < 1$.
> > > > >
> > > > > The updated submission includes explicit assumptions on $\varepsilon$ in the SCO theorems.
> > > > >
> > > > > > For the composition theorem, I would really appreciate it if the authors can point me to the exact place where things differ and where adapting Dwork et al. (or many other works following that) would not work. I am just trying to understand the paper and want to end this review process at the stage where if I accept the paper, I can defend it as my own. For example, the dependence of different $\varepsilon_1, \ldots, \varepsilon_k$ is mentioned implicitly in John Duchi's notes (https://web.stanford.edu/class/stats311/lecture-notes.pdf) and Corollary 2.3 in this paper (https://arxiv.org/pdf/1912.04042.pdf).
> > > > >
> > > > > We think the following explanation may resolve the confusion. The previous submission draft had two slightly different statements of Lemma 3.3. The incorrect statement in the main body assumed a divergence property that holds for all pairs of neighboring datasets; the correct statement in the Appendix (and used in the overall proof) only assumed the divergence property for a fixed pair of neighboring datasets. We apologize for this error and have updated the submission draft.
> > > > >
> > > > > The difference between these statements is the key reason our argument cannot just swap in existing statements for advanced composition. Inside the proof of Theorem 3.2, we fix some pair of neighboring datasets $X$ and $X'$ and then use Lemma 3.1 to bound the divergence between each coordinate output in $X$ and $X'$ in terms of the actual difference between $X$ and $X'$ (see the $a_j / 2\Delta_2$ term). Then we apply Lemma 3.3 to combine the individual coordinate guarantees into an overall divergence guarantee, and finally a differential privacy guarantee. In particular, we cannot replace Lemma 3.3 with a generic advanced composition result, even one with heterogeneous privacy parameters, because such guarantees lack the specific dependence on the distance between a fixed pair of datasets, and the proof relies on this dependence. In other words, we require a slightly more fine-grained version of these advanced composition guarantees, and that is Lemma 3.3.
> > > > >
> > > > > That said, while our specific statement of Lemma 3.3 is necessary for our argument, we agree that the actual proof of Lemma 3.3 is not very different from other proofs of advanced composition. We have updated the text before Lemma 3.3 in an attempt to clarify this and apologize for any imprecision in our previous statements.

---

> > > > > > ### Comment · Reviewer_Frn5 · 2021-11-20
> > > > > > **re: discussion**
> > > > > >
> > > > > > "Perhaps there's some confusion over which Bassily et al. paper the ICML paper is referring to? The ICML paper emphasizes that its SCO results match those of Bassily, Smith, Thakurta 14 (arxiv.org/abs/1405.7085)."
> > > > > >
> > > > > > I see. Yes, I know that BST14 required the use of exponential mechanism to move from ERM to population risk. Thanks for clearing that ICML paper only has the result matching BST14 not BFTT19. Perhaps, it might be a good idea to keep both results in the final table just to give a clarity to the readers.
> > > > > >
> > > > > > The incorrect statement in the main body assumed a divergence property that holds for all pairs of neighboring datasets; the correct statement in the Appendix (and used in the overall proof) only assumed the divergence property for a fixed pair of neighboring datasets.
> > > > > >
> > > > > > This makes much more sense. Thanks a lot!
> > > > > >
> > > > > > I think the paper seems good for me then. My suggestion based on our discussion so far would be as follows (some of which the authors have already incorporated):
> > > > > > -- Make a clear distinction with the binomial mechanism of Dwork et al. I was reading the EUROCRYPT21 paper by Ghazi et al. yesterday and some of the ideas seem close to this paper. It might be a good idea to clearly distinguish the technical as well as algorithmic aspects of these three papers when discussing the scalar summation algorithm and its analysis.
> > > > > > -- Maybe, make it explicit that one open problem would be to reduce the communication complexity of your protocol while keeping the simplicity intact so that anyone who wants to look for an open problem do not have to rediscover this entire discussion.
> > > > > > -- Write the correct form of composition lemma and explicitly state the difference with the previous ones. Why the previous one does not work and why you have to give self-contained proof.
> > > > > > -- Give the exact dependence on privacy budge in the main theorem and use $\epsilon <1$ to give a corollary. This is mostly for any practitioner who wants to explore this idea in the future.
> > > > > > -- On my second reading, the interactive protocol seems to be the impressive part. I would suggest having a note about it from the very start. Discuss the pros and cons of the interaction (where many papers starting with Smith et al. have discussed it), keep an honest discussion with one pass practical paper with respect to convexity requirement for privacy proof, multiple pass algorithm, etc, etc.
> > > > > > -- I noticed one result where the authors cite Duchi et al. with the citation to their conference paper. The paper is now in a journal and journal citation should be given more weightage. Also, DJW gives lower bound in $\epsilon$-LDP setting. Since the authors have interactivity, the reduction given in Bun et al. local heavy hitters would not work here. So, maybe, make this comparison clear as well.
> > > > > >
> > > > > > In general, I think the paper would benefit a lot from adding a section on the technical aspects of the paper. I am reading this paper as a theory paper. So, it is a good idea to make it as precise all these statements as possible.
> > > > > >
> > > > > > Finally, thanks to the authors to clarify many of my doubts. Once the authors have made these (minor) changes, I would update the score to accept.

---

> > > > > > > ### Author Response · Authors · 2021-11-23
> > > > > > > **re: discussion**
> > > > > > >
> > > > > > > > Thanks for clearing that ICML paper only has the result matching BST14 not BFTT19. Perhaps, it might be a good idea to keep both results in the final table just to give a clarity to the readers.
> > > > > > >
> > > > > > > We added a short discussion of the ICML one-pass paper that highlights its weaker SCO guarantee to try to clarify this. See response to "On my second reading … " question below for details.
> > > > > > >
> > > > > > > > Make a clear distinction with the binomial mechanism of Dwork et al. I was reading the EUROCRYPT21 paper by Ghazi et al. yesterday and some of the ideas seem close to this paper. It might be a good idea to clearly distinguish the technical as well as algorithmic aspects of these three papers ...
> > > > > > >
> > > > > > > The latest submission modifies the paragraph after Lemma 3.1 with the following sentences: "This sum has binomial noise, which was first analyzed by Dwork et al. (2006a) in the context of central DP. We adapt work on a shuffle private variant due to Ghazi et al. (2021) to incorporate a dependence on the per-instance distance between databases." We also added similar language before the proof of Lemma A.1 in the Appendix.
> > > > > > >
> > > > > > > > Maybe, make it explicit that one open problem would be to reduce the communication complexity of your protocol while keeping the simplicity intact
> > > > > > >
> > > > > > > Added the following sentence to the end of Section 3.1: "Improving the communication efficiency of these protocols may be an interesting direction for future work."
> > > > > > >
> > > > > > > > Write the correct form of composition lemma and explicitly state the difference with the previous ones. Why the previous one does not work and why you have to give self-contained proof.
> > > > > > >
> > > > > > > See Lemma 3.3 and the preceding paragraph.
> > > > > > >
> > > > > > > > Give the exact dependence on privacy budget in the main theorem and use $\varepsilon < 1$  to give a corollary. This is mostly for any practitioner who wants to explore this idea in the future.
> > > > > > >
> > > > > > > On closer inspection, we can replace our assumption $\varepsilon < 1$ with a much more generous $\varepsilon \leq 15$. This comes into play in the following two places in our analysis: 1) proving $\alpha m p \geq 2t$ in the proof of Lemma A.2, and 2) bounding divergence by $\varepsilon$ near the end of the proof of Theorem A.6. This modification requires the addition of two very mild assumptions: we now additionally generally assume $\delta < 1/2$, and set $g \geq 4$ in the proof of Lemma A.6.
> > > > > > >
> > > > > > > > On my second reading, the interactive protocol seems to be the impressive part. I would suggest having a note about it from the very start. Discuss the pros and cons of the interaction (where many papers starting with Smith et al. have discussed it), keep an honest discussion with one pass practical paper with respect to convexity requirement for privacy proof, multiple pass algorithm, etc, etc.
> > > > > > >
> > > > > > > The previous update added discussions of interaction at the beginning of Section 1.1 and around Definition 2.8. We have added the following discussion of the one-pass paper to Section 1.2: "KMSTTX21 study DP-FTRL, whichuses a single pass, extends to nonconvex losses, does not rely on shuffling or amplification, and satisfies central DP. It is possible to adapt their algorithm to sequentially interactive shuffle privacy using our vector sum protocol, though their $O(d^{1/4}/\sqrt{n})$ SCO guarantee is weaker than our $O(d^{1/3}/n^{2/3})$ SCO guarantee (Theorem 4.7)."
> > > > > > >
> > > > > > > > [Duchi et al. 13] is now in a journal
> > > > > > >
> > > > > > > The journal version was published 5 years after the conference version, so we feel that citing it confuses the actual chronology of results. As a result, we prefer to cite the conference version, as this best preserves when the relevant ideas appeared in the literature. However, if it seems necessary, the newest submission update cites the journal version instead.
> > > > > > >
> > > > > > > > DJW gives lower bound in $\varepsilon$-LDP setting. Since the authors have interactivity, the reduction given in Bun et al. local heavy hitters would not work here.
> > > > > > >
> > > > > > > The local heavy hitters result does extend DJW's pure result to approximate DP (see e.g. Lemma 5.2 in arxiv.org/pdf/1904.03564.pdf) in the sequentially interactive (but not fully interactive) case. The previous updated added some language to clarify sequential vs fully interactive locally private lower bounds at the end of the second paragraph of the introduction.
> > > > > > >
> > > > > > > > In general, I think the paper would benefit a lot from adding a section on the technical aspects of the paper
> > > > > > >
> > > > > > > We hope the changes described above help clarify some of the paper's technical aspects. Adding a new section is tricky given the ICLR space constraints and the text already added, but we will keep this in mind when updating the Arxiv version.
> > > > > > >
> > > > > > > Finally, thanks for all of the effort you've put into the review process! We feel that these exchanges have improved the paper.

---

> > > > > > > > ### Comment · Reviewer_Frn5 · 2021-11-23
> > > > > > > > **re: discussion**
> > > > > > > >
> > > > > > > > Thanks to the authors for the painstaking clarification regarding the different aspects of the paper. I have a much better understanding of the paper now.
> > > > > > > >
> > > > > > > > exact dependence on privacy budget in the main theorem:
> > > > > > > > I did some calculations on my own today and was about to suggest that it seems possible that the dependence can be improved to the most practical setting of $\epsilon$ modulo something that I might have overlooked. I did not get $\epsilon <15$, but rather $\epsilon \leqslant 10$ and I was pretty happy with it, too. If the authors have good reasons to believe that they can get to $\epsilon $ up to $15$, I think they should add that comment (at least in the camera-ready if the paper gets finally accepted).
> > > > > > > >
> > > > > > > > On journal vs conference: this is more of my personal opinion that a journal paper should be cited (it is often better reviewed than conference) or at least both of them should be cited if a journal version already exists.

---

### Decision · Program_Chairs · 2022-01-20

**Decision:**

Accept (Poster)

**Comment:**

This work is on stochastic convex optimization (SCO) in shuffle differential privacy (DP) models. In SCO, a learner receives a convex loss function L: Theta x X -> Reals, where Theta is a d-dimensional vector of parameters and X is a set of data points. The objective is to use samples x1, x2, …, xn to find a parameter theta that minimizes the loss E_{x ~ D}[L(theta,x)], where the distribution D on X is unknown. The shuffle models considered are a ``sequential" model where the analyzer operates in rounds (and where a new set of users participate in a local DP protocol in every round), and a new, stronger "full" model in which the analyzer can request a specific subset of users to participate in a round, which in particular allows users' data to be queried more than once. This work shows that in the full model, one can develop excess population loss bounds matching the known best-possible bounds in centralized DP; it is also shown that even the weaker sequential model offers improved excess population loss bounds over the best-possible bound of sqrt(d/n) in the local setting.

The reviewers appreciated the novelty and technical depth of this work (despite concerns about part of the work being taking “off the shelf” results).